# Numerical Verification of the Drive-By Monitoring Method for Identifying Vehicle and Bridge Mechanical Parameters

Kyosuke Yamamoto [1,*], Ryota Shin [2] and Eugene Mudahemuka [3]

1   Institute of Systems and Information Engineering/Center for Artificial Intelligence Research, University of Tsukuba, Tsukuba 305-8573, Japan
2   Doctoral Program in Policy and Planning Sciences, Graduate School of Science and Technology, University of Tsukuba, Tsukuba 305-8577, Japan
3   Degree Program in Engineering Mechanics & Energy, Graduate School of Science and Technology, University of Tsukuba, Tsukuba 305-8573, Japan
*   Correspondence: yamamoto.kyosuke.fu@u.tsukuba.ac.jp; Tel.: +81-29-853-5146

**Featured Application: Using the PRE method, the mechanical parameters (i.e., mass, damping, stiffness) of the vehicle–bridge interaction system and road unevenness can be simultaneously estimated only from vehicle vibration and position data. In this paper, it is clarified by numerical simulation that stiffness reductions of the bridge model can be identified by the PRE method. This result has increased the feasibility of bridge inspection based on vehicle vibration.**

**Abstract:** The PRE (numerical simulation-based vehicle and bridge parameter and road roughness estimation) method uses vehicle vibration data to identify the vehicle's and bridge's mechanical parameters and estimate road unevenness simultaneously. This method randomly assumes the mechanical parameters first. Secondly, it solves the vehicle's IEP (input estimation problem) and the bridge's DRS (dynamic response simulation) from the vehicle vibration data to obtain road profiles of the front and rear wheels. Repeat the random assumption of the mechanical parameters to minimize the residual between the obtained road unevenness because the road unevenness of the front and rear wheels are expected to match. To search for a better combination of the mechanical parameters, the MCMC (Monte Carlo Markov chain) algorithm is adopted in this paper. This paper also numerically simulates vehicle vibration data for the cases of the reduced-stiffness bridge model and examines whether this method can identify the position, range, and magnitude of stiffness reduction. The numerical simulation results show that bridge-stiffness reduction can be estimated reasonably.

**Keywords:** vehicle–bridge interaction system; system identification; road unevenness; bridge inspection; drive-by monitoring; the PRE method

## 1. Introduction

The safety of existing road bridges has been a priority issue in engineering for decades because none can deny that bridge deterioration is a real threat to our lives and economies. This safety problem is mainly a concern among bridge owners and investors who are worried about the rapid deterioration of aged bridges if not routinely inspected and treated appropriately. In addition, the number of skilled engineers who can appropriately inspect bridges is still limited. Since resources such as engineers and budgets are limited, it is necessary to classify (1) bridges that should be repaired as soon as possible and (2) bridges that are damaged but should be just monitored without immediate repair. For the latter, developing a system to detect the damage progress at an early stage is necessary. An option as a specific method for realizing early damage detection is vibration-based SHM (structural health monitoring) [1–4].

In vibration-based SHM, sensors are usually installed on a bridge [3,4]. Measured vibration data is often used to extract the natural frequencies [5] and other modal pa-

rameters [6]. Modal parameters are fundamental properties of a structure that reflect its structural condition [2]. When a structure experiences damage, these parameters may shift [7] because the stiffness decreases. Identifying such changes promptly and precisely is crucial for maintaining the system's structural integrity.

There are many signal-processing methods for determining structural models from collected data. Structures are often considered a system. In this idea, forces acting on the structure are inputs, while vibration responses are outputs. The system parameters are mechanical parameters, such as mass, damping, and stiffness of the structure. In SHM, estimating these parameters only from the outputs is necessary. If the inputs are wind- and traffic-induced vibrations, they are often treated as white noise [6,8,9] to make system identification based on the output data alone. Examples of parametric methods in the time domain include the AR (autoregressive) [10] and SSI (stochastic subspace identification) methods [11]. On the other hand, the FDD (frequency domain decomposition) method [12] derives a singular value spectrum through the SVD (singular value decomposition) of the cross-power spectrum matrix of the multipoint responses. These schemes provide all information about the system parameters of the target structure.

Vibration-based SHM for bridges used to be intended mainly for constantly monitoring large-span bridges instead of routine inspections [2–4]. Thus, if applying SHM to many aged short-span bridges widely distributed in local areas, the cost of installing sensors on each bridge would make the SHM implementation difficult.

The applicability of this direct approach to the large numbers of old bridges poses various feasibility concerns related to bridge location, sensor affordability, and installation conditions. Another drawback to the traditional SHM method is the absence of information related to the intact version of the target bridge. These problems call for an alternative bridge monitoring method that is easy to be implemented, cost-effective, able to estimate current performance, and applicable to short-span bridges. One option to be expected to overcome these issues is to apply drive-by bridge monitoring technology. Drive-by bridge monitoring uses vehicle vibration data to evaluate bridge health conditions, as shown in Figure 1. This kind of technique has been investigated for decades [13,14].

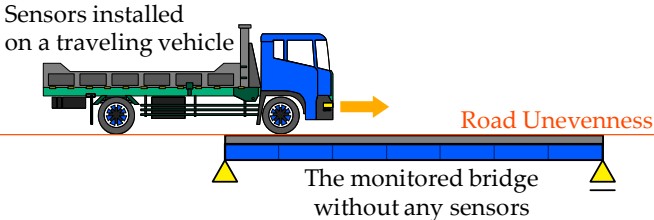

**Figure 1.** The conceptual diagram of the drive-by bridge monitoring: The measured vibration data on a traveling vehicle is used to evaluate the bridge's health. In this study, bridges are modeled as simple Euler–Bernoulli beam supported by pins.

The most known in this regard is the indirect approach proposed for the first time by Yang et al. [15]. This method uses only vehicle vibration to estimate the bridge's natural frequency. They apply the Fourier transform to the vehicle acceleration response and extract the bridge frequency. The obtained spectrum of their study has three predominant frequencies: vehicle drive frequency, two shifted bridge frequencies, and the vehicle's natural frequency. This idea has also been verified experimentally [16]. However, the approach mentioned above is only a success for the first natural frequency of the bridge. On the other hand, it is known that higher modal parameters are more sensitive to local stiffness changes. Thus, to extend this idea to find higher modal parameters, Yang and Chang [17] have enhanced this idea with the EMD (empirical mode decomposition) method to identify up to the seventh natural frequency of the bridge.

The indirect approach inspires many researchers: Xiang et al. [18] added a vibration exciter on a traveling vehicle to improve the estimation accuracy. The authors [19] also examined the feasibility of estimating bridge mode shapes from the vibration responses

of several cargo trailers. The other frequency-based approaches [20] are made by Gonzalez et al. [21] and Wang et al. [22], while the mode shape-based approaches are also studied by Yang et al. [23] and Malekjafarian et al. [24].

Gonzalez et al. [21] have verified the drive-by monitoring method for estimating the bridge's natural frequencies. They validate their findings by performing the tests over different running speeds, road profiles, and traffic conditions. The acceleration responses of a vehicle crossing the bridge are fitted with an accelerometer and a GPS device. In their experiment, the first natural frequency cannot be detected, while the second natural frequency is detected at the low speed of the monitoring vehicle. This inability to detect the natural frequency is attributed to the high speeds, the absence of heavy traffic, and the interference of road profile frequencies. This research generally concludes that the accuracy of extracted natural frequency becomes better as the bridge length increases, vehicle speed decreases, and the road profile is smooth. A similar trend can also be confirmed in the study of Yang et al. [23]. Identifying the bridge's natural frequency is difficult because vehicle frequency and road roughness interfere with bridge frequencies. This issue has been addressed by Yang et al. [20]. They use vehicle–bridge contact response, which eliminates vehicle frequencies, and a shaker fixed on the bridge to enhance bridge vibration. Applying these ideas could reduce road roughness frequencies from the obtained data. It has concluded that this combination is effective for general application since the shaker's frequency and location can be tuned up by dynamic amplification factor to meet various bridges.

On the other hand, Yang et al. [23] use a one-tractor-two-trailers system and measure the vibration data of the two trailers. The tractor is a shaker traveling on the bridge, while the trailers less susceptible to engine vibration are the observers of bridge vibrations in this scheme. Using residual vibration between two trailers can reduce the road roughness frequencies from the obtained data and extract the bridge vibration components. From the extracted data, bridge mode shapes can be estimated accurately.

According to these previous studies, the feasibility of drive-by bridge monitoring has been increased. Kim et al. [25] also show the applicability of this technology to short-span bridges, which means it can provide a solution to the vast demand for bridge inspections.

As mentioned above, many drive-by monitoring approaches have estimated modal parameters under carefully limited conditions and have evaluated bridge health. Feature-value-based bridge health monitoring is associated with some challenges. When using features such as modal parameters for detecting and estimating bridge damages in vibration-based SHM, it is also necessary to verify not only their estimation accuracy but also the sensitivity of focused modal parameters to target damages. Since drive-by monitoring is an indirect approach [13], estimated modal parameters only suggest the possibility of bridge damages and never identify the damages. Furthermore, there is no information about the dynamic characteristics of the intact state of old bridges because the aging bridges have been in service for a long time. In most cases, it is identified that the first vibration mode is sensitive only to global stiffness reduction, while the extraction of a higher mode sensitive to local stiffness reduction is still a challenge. Due to the limitation of feature-value-based monitoring, an alternative concept is required for bridge health screening. Thus, the implementation of effective monitoring on many bridges should be based on a direct evaluation of the bridge's structural performance. This implies that it is preferable to estimate and evaluate bridge stiffness directly.

Zhao et al. [26] have proposed a method to estimate road profiles from vehicle vibration data without prior information about the vehicle's mechanical parameters. This method can identify the vehicle's mechanical parameters during estimating road unevenness. Keenahan et al. [27], Hasegawa et al. [28], and Xue et al. [29] have studied similar methods. While Keenahan et al. [27] and Hasegawa et al. [28] have examined their schemes only numerically, Zhao et al. [26] and Xue et al. [29] have verified the applicability of their method by using actual data. According to the experimental result, Zhao's method can accurately estimate road profiles. In the field experiment, Zhao et al. [26] use a known size

hump to decide the initial distribution of the vehicle parameters and update them by GA (Genetic Algorithm). Keenahan et al. [27] extend Zhao's method to use randomly-assumed initial parameters instead of using calibration data. In this study, while the prior probability distribution of each vehicle parameter is assumed to be the Gaussian distribution, the posterior distribution accurately indicates each correct value. Xue et al. [29] also apply a random method to the actual vehicle vibration data for simultaneous vehicle identification and road profile estimation. The Monte Carlo-based schemes [27,29] show high accuracy in estimating road profiles and are verified numerically and experimentally to be applicable. Furthermore, the obtained high accuracies have increased momentum to extend the Monte Carlo-based schemes to bridge inspection. However, it is still necessary to improve the accuracy of extracting bridge responses as well as road profiles from vehicle vibrations. Hasegawa et al. [28] have tried to avoid the numerical integration process by introducing contact forces into the state variable vector of a vehicle model. The motivation of these existing studies [27,28] makes an input estimation process more accurate for extracting bridge information.

Therefore, the authors have extended their methods [26–29] and have proposed a novel scheme that can simultaneously estimate not only vehicle parameters but also bridge parameters. This scheme extension can be realized by adding a bridge simulation process. The extended method is called the PRE (numerical simulation-based vehicle and bridge parameter and road roughness estimation in VBI system) method [30–32] in this study. The uniqueness of the PRE method is that it includes the identification process of the bridge system.

The proposed PRE method has a different purpose from similar previous studies. For instance, the publications [26–29] use PRE-like schemes to estimate road unevenness, unlike the proposed method, which estimates bridge parameters as well as road unevenness. Therefore, the required accuracy of the proposed method for estimating vehicle inputs is much higher.

The PRE method is probably the world's first drive-by scheme for simultaneously estimating vehicles, bridges, and road unevenness. With this scheme, all information about the VBI (vehicle–bridge interaction) system can be reproduced only from vehicle vibration and position data. However, the applicability of this method to bridge inspection has not been thoroughly investigated. In the general bridge inspections, changes in appearance and hammering sounds are collected mainly to estimate stiffness changes. Thus, the parametric changes in bridge stiffness should be methodologically considered in this paper.

The vehicle's vertical acceleration vibrations and position information are measured in the PRE method. These measured data and assumed mechanical parameters are used to estimate two road profiles of the front and rear wheels in the process. Both estimated road surface unevenness functions should match when the vehicle moves straight ahead. However, the estimated road unevenness in the initial step rarely matches because the mechanical parameters are randomly assumed. Therefore, the sum of the squared error between the road unevenness estimated at the front and rear wheels is adopted as the objective function to find the optimal mechanical parameters. The PRE method searches for the mechanical parameters minimizing this function. It is noted that the wheelbase and gross weight of the vehicle are assumed to be known because the obtained solution could be indeterminate. Because these two known parameters are easy to be measured, the PRE efficiency does not decrease. According to the previous studies of the PRE method [30,31], the converged parameters can match the correct values. The same tendency can be confirmed in road profile estimation using the random parameter assumption [27].

There are several methods of searching for the optimal solution of the mechanical parameters. Zhao et al. [26] adopted GA to identify the parameters, while Keenahan et al. [27] applied cross-entropy optimization. The success of these existing similar studies indicates that a scheme based on Monte Carlo simulation can be applied to the PRE method. At the same time, it should be noted that the difference between this paper and the existing studies [26–29] lies in bridge system identification. The PRE method needs to search for the

optimal solution in an ample unknown multidimensional space. Thus, as the first step, this paper adopts the MCMC (Monte Carlo Markov chain) model's so-called random walk for the optimization process.

The damage detection performance of the PRE method should be evaluated quantitatively to verify its applicability. In previous studies [30,31], structural changes have been mainly considered as stiffness reduction; according to the findings of these studies, the PRE method can estimate the location and magnitude of stiffness reduction. However, whether the stiffness reduction can be detected in places with slight vibration, such as at a bridge end, is unclear. Furthermore, the estimation sensitivity to a slight decrease in stiffness should be investigated. Another current study [32] provides the application results of the PRE method to the data obtained from the field experiment and concludes that there are still many difficulties in implementing this scheme using actual data.

Therefore, this study aims to verify the proof-of-concept of the MCMC-powered PRE method for estimating bridge stiffness reduction. The dynamic responses of the VBI system are numerically simulated while changing the bending stiffness of the bridge model. Twelve patterns with different locations, sizes, and magnitudes of stiffness reduction are prepared. Since the MCMC process gives the estimation results as probability density distributions, the means and standard deviations of the estimated parameters are used as indices to quantitatively express the accuracy and reliability of the PRE method, respectively.

## 2. Numerical Simulation

This section introduces the basic theory of the VBI system. The vehicle vibration accelerations are simulated as measured data. While the vehicle is modeled by one rigid body and two mass points with springs and dashpots, the bridge is modeled by the FEM (finite element method) using the Hermite basis.

### 2.1. The Basic Theory of the Vehicle–Bridge Interaction System

The VBI system consists of two sub-systems: a vehicle system and a bridge system. The vehicle system's inputs are the road unevenness and the bridge vibration components, while its output is the vehicle vibration. On the other hand, the bridge system's input is the contact force due to the vehicle vibration, while its output is the bridge vibration. The outputs of one system are the inputs of the opposite system. Therefore, the vehicle and bridge form an interaction system. The conceptual diagram of this system is shown in Figure 2.

First, the equations of motion of the vehicle are given by

$$\mathbf{M}_v \ddot{\boldsymbol{z}}(t) + \mathbf{C}_v \dot{\boldsymbol{z}}(t) + \mathbf{K}_v \boldsymbol{z}(t) = \boldsymbol{F}_v(t) \tag{1}$$

where

$$\mathbf{M}_v = \begin{bmatrix} \frac{d_2 m_s}{d_1+d_2} & \frac{d_1 m_s}{d_1+d_2} & 0 & 0 \\ \frac{I}{d_1+d_2} & \frac{I}{d_1+d_2} & 0 & 0 \\ 0 & 0 & m_{u1} & 0 \\ 0 & 0 & 0 & m_{u2} \end{bmatrix} = \begin{bmatrix} m_{s1} & m_{s2} & 0 & 0 \\ \frac{I}{d_1+d_2} & \frac{I}{d_1+d_2} & 0 & 0 \\ 0 & 0 & m_{u1} & 0 \\ 0 & 0 & 0 & m_{u2} \end{bmatrix} \tag{2}$$

$$\mathbf{C}_v = \begin{bmatrix} c_{s1} & c_{s2} & -c_{s1} & -c_{s2} \\ d_1 c_{s1} & -d_2 c_{s2} & -d_1 c_{s1} & d_2 c_{s2} \\ -c_{s1} & 0 & c_{s1} & 0 \\ 0 & -c_{s2} & 0 & c_{s2} \end{bmatrix} \tag{3}$$

$$\mathbf{K}_v = \begin{bmatrix} k_{s1} & k_{s2} & -k_{s1} & -k_{s2} \\ d_1 k_{s1} & -d_2 k_{s2} & -d_1 k_{s1} & d_2 k_{s2} \\ -k_{s1} & 0 & k_{s1} + k_{u1} & 0 \\ 0 & -k_{s2} & 0 & k_{s2} + k_{u2} \end{bmatrix} \tag{4}$$

$$\boldsymbol{z}(t) = \begin{bmatrix} z_{s1}(t) & z_{s2}(t) & z_{u1}(t) & z_{u2}(t) \end{bmatrix}^T \tag{5}$$

$$\boldsymbol{F}_v(t) = \begin{bmatrix} 0 & 0 & k_{u1}u_1(t) & k_{u2}u_2(t) \end{bmatrix}^T \tag{6}$$

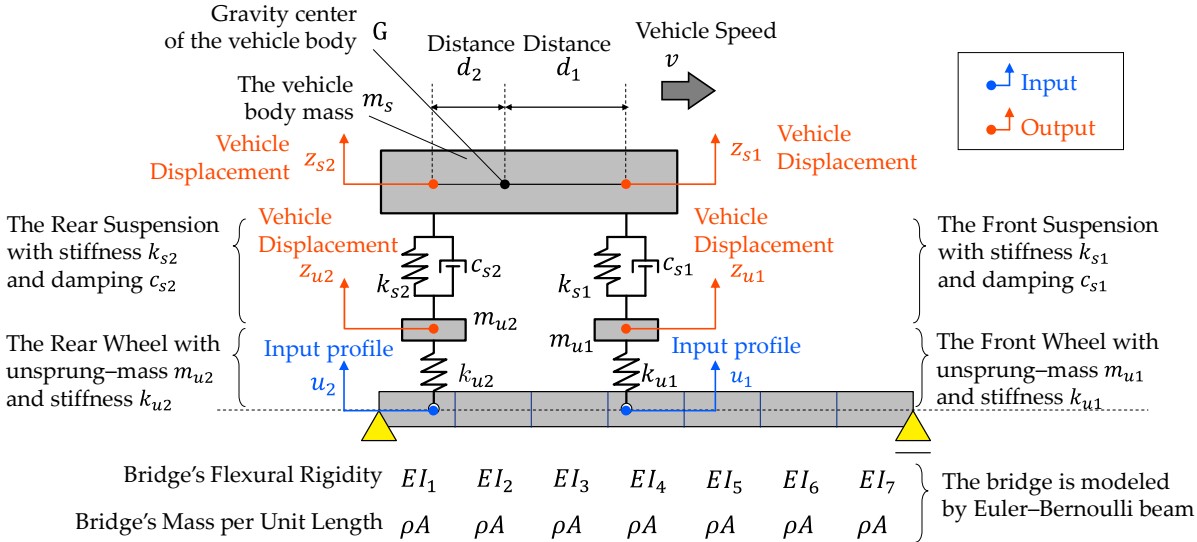

**Figure 2.** The adopted model and mechanical parameters of the VBI (vehicle–bridge interaction) system: The vehicle is assumed as an RBSM (rigid-body-spring model), while the bridge model is based on the Euler–Bernoulli theorem.

The vertical vibration displacements of sprung-mass and unsprung-mass at the $i$-th wheel are $z_{si}(t)$ and $z_{ui}(t)$, respectively. The operators $(\dot{\ })$ and $(\ddot{\ })$ represent the first- and second-order time derivatives. The measured data is $\ddot{z}(t)$. Let $d_i$ be the distance from the gravity point G to the $i$-th suspension. The mass of the vehicle body is denoted by $m_s$. The inertia of the vehicle body can be given by $I = m_s d_1 d_2$. The damping $c_{si}$ and spring stiffness $k_{si}$ characterize the $i$-th suspension. The mass and stiffness of the tires are $m_{ui}$ and $k_{ui}$. The forced vertical displacement under the $i$-th wheel is called the input profile $u_i(t)$ in this paper. Let $\mathbf{M}_v$, $\mathbf{C}_v$, and $\mathbf{K}_v$ be the vehicle mass, damping, and stiffness matrices, respectively. $\boldsymbol{F}_v$ represents the external force vector acting on the vehicle system. This vehicle model is generally called a half-car model.

The bridge is modeled as an Euler–Bernoulli beam and, therefore, $\rho A(x)$ and $EI(x)$ denote the mass per unit length and flexural rigidity of the beam, respectively. Possible aging-caused changes in the bridge system are modeled as the reduction of the bending stiffness $EI(x)$ in the numerical simulations. The equation of motion of the bridge can be expressed as

$$\rho A \ddot{y}(x,\ t) + \frac{\partial^2}{\partial x^2} EI\left(\frac{\partial^2}{\partial x^2} y(x,\ t)\right) = \sum_{i=1}^{2} \delta(x - x_i(t)) P_i(t) + \delta(x) R_{\mathrm{A}}(t) + \delta(x - L) R_{\mathrm{B}}(t) \tag{7}$$

where $y(x,\ t)$ is the bridge vibration, while $P_i(t)$ denotes the contact force of $i$-th tire. $\delta(x)$ is Dirac's delta function. $R_{\mathrm{A}}$ and $R_{\mathrm{B}}$ are the reaction forces acting at the supports. The position of $i$-th wheel is given by $x_i(t)$.

To simulate the dynamic responses of the bridge, FEM is adopted in this study. Assuming that the average of the weighted residual is zero, the given equation becomes

$$\int_0^L \omega \left(\rho A \frac{\partial^2 y}{\partial t^2} + EI \frac{\partial^4 y}{\partial x^4} - p\right) \mathrm{d}x = 0 \tag{8}$$

where $\omega$ is the weight function, and $p$ summarizes the right-hand side terms of Equation (7). The weak form of this equation is expressed as

$$\int_0^L \omega \rho A \frac{\partial^2 y}{\partial t^2} \mathrm{d}x + \int_0^L \frac{\partial^2 \omega}{\partial x^2} EI \frac{\partial^2 y}{\partial x^2} \mathrm{d}x = \int_0^L \omega p \, \mathrm{d}x \tag{9}$$

Let $\mathbf{N}(x)$ and $\mathbf{y}(t)$ be the Hermite interpolation basis functions and numerical solutions of bridge vibration displacements. The approximate solution can be given by $y(x, t) \approx \mathbf{N}(x) \cdot \mathbf{y}(t)$. Now, assuming that the weight function is also approximated by $\omega(x) \approx \mathbf{N}(x) \cdot \boldsymbol{\omega}$, this is called the Galerkin method, which is the most popular scheme for FEM. Substituting both approximated variables into Equation (9), the equation becomes

$$\boldsymbol{\omega}^{\mathrm{T}} \big( \mathbf{M}_b \ddot{\mathbf{y}}(t) + \mathbf{K}_b \mathbf{y}(t) - \mathbf{F}_b(t) \big) = 0 \tag{10}$$

where the obtained matrices are $\mathbf{M}_b = \int_0^L \mathbf{N}\mathbf{N}^{\mathrm{T}}\mathrm{d}x$ (the global mass matrix) and $\mathbf{K}_b = \int_0^L \frac{\partial^2 \mathbf{N}}{\partial x^2} \frac{\partial^2 \mathbf{N}}{\partial x^2}^{\mathrm{T}} \mathrm{d}x$ (global stiffness matrix), respectively. The external force vector is

$$\begin{aligned} \mathbf{F}_b(t) &= \int_0^L \mathbf{N}(x) p(x, t) \mathrm{d}x \\ &= \mathbf{N}(x_1(t)) P_1(t) + \mathbf{N}(x_2(t)) P_2(t) + \mathbf{N}(0) R_{\mathrm{A}}(t) + \mathbf{N}(L) R_{\mathrm{B}}(t) \\ &= \mathbf{L}(t) \left\{ \begin{array}{c} P_1(t) \\ P_2(t) \end{array} \right\} + \mathbf{H}(t) = \mathbf{L}(t)\mathbf{P}(t) + \mathbf{H}(t) \end{aligned} \tag{11}$$

where $\mathbf{L}(t) = \begin{bmatrix} \mathbf{N}(x_1(t)) & \mathbf{N}(x_2(t)) \end{bmatrix}$, and $\mathbf{H}(t)$ is the reaction force vector. When $g$ denotes the gravity acceleration ($g = -9.8 \text{ m/s}^2$), the contact forces acting on the bridge are

$$P_1(t) = \frac{d_2 m_s}{d_1 + d_2}(g - \ddot{z}_{s1}) + m_{u1}(g - \ddot{z}_{u1})$$

$$P_2(t) = \frac{d_1 m_s}{d_1 + d_2}(g - \ddot{z}_{s2}) + m_{u2}(g - \ddot{z}_{u2}) \tag{12}$$

The position of the contact force changes when the wheel position changes as the vehicle's travel. Then, assuming that Equation (10) is always zero for arbitrary $\omega(x, t)$, and introducing the damping term, the following equation to be solved is obtained:

$$\mathbf{M}_b \ddot{\mathbf{y}}(t) + \mathbf{C}_b \dot{\mathbf{y}}(t) + \mathbf{K}_b \mathbf{y}(t) = \mathbf{F}_b(t) \tag{13}$$

where the damping is assumed to be $\mathbf{C}_b = \alpha \mathbf{M}_b + \beta \mathbf{K}_b$. $\alpha$ and $\beta$ are called the coefficients of Rayleigh damping.

The interactions of this system are represented by the input profiles $\mathbf{u}(t)$ and the contact forces $\mathbf{P}(t)$. The input profiles consist of the road profiles $\mathbf{r}(t)$ and the bridge vibration components $\widetilde{\mathbf{y}}(t)$:

$$\mathbf{u}(t) = \mathbf{r}(t) + \widetilde{\mathbf{y}}(t) \tag{14}$$

The road profile of $i$-th wheel $r_i(t)$ is the vector component of $\mathbf{r}(t)$. This variable is represented as a time function, but it is originally derived from a spatial function $R(x)$, which denotes road unevenness. The vehicle traveling effect transforms road unevenness into a time function, as shown in the following formula:

$$r_i(t) = R(x_i(t)) \tag{15}$$

The bridge vibration component $\widetilde{\mathbf{y}}(t)$ is also derived by considering the traveling effect of the vehicle. The bridge deflection input to the vehicle is originally a time-space function $y(x, t)$. However, as the location $x$ becomes a vehicular wheel position $x_i(t)$, which is a

time function, the bridge deflection is transformed from a time-space function $y(x, t)$ to a time function $\widetilde{y}_i(t)$, as shown in the following formula:

$$\widetilde{y}(t) = \begin{Bmatrix} y(x_1(t), t) \\ y(x_2(t), t) \end{Bmatrix} = \begin{Bmatrix} \mathbf{N}(x_1(t)) \cdot \mathbf{y}(t) \\ \mathbf{N}(x_2(t)) \cdot \mathbf{y}(t) \end{Bmatrix} = \mathbf{L}(t)^{\mathrm{T}} \mathbf{y}(t) \tag{16}$$

By integrating all formulas above, the equation of motion of the VBI system can be expressed by the following formula:

$$\begin{bmatrix} \mathbf{M}_v & \mathbf{O} \\ \mathbf{L}^{\mathrm{T}}(t)\mathbf{M}_p & \mathbf{M}_b \end{bmatrix} \begin{Bmatrix} \ddot{z}(t) \\ \ddot{y}(t) \end{Bmatrix} + \begin{bmatrix} \mathbf{C}_v & \mathbf{O} \\ \mathbf{O} & \mathbf{C}_b \end{bmatrix} \begin{Bmatrix} \dot{z}(t) \\ \dot{y}(t) \end{Bmatrix} + \begin{bmatrix} \mathbf{K}_v & -\mathbf{K}_p\mathbf{L}^{\mathrm{T}}(t) \\ \mathbf{O} & \mathbf{K}_b \end{bmatrix} \begin{Bmatrix} z(t) \\ y(t) \end{Bmatrix} = \begin{Bmatrix} \mathbf{K}_p r(t) \\ \mathbf{L}(t)\mathbf{M}_v g + R(t) \end{Bmatrix} \tag{17}$$

where $\mathbf{O}$ represents a zero matrix, and

$$\mathbf{K}_p = \begin{bmatrix} 0 & 0 \\ 0 & 0 \\ k_{u1} & 0 \\ 0 & k_{u2} \end{bmatrix} \tag{18}$$

$$\mathbf{M}_p = \begin{bmatrix} \frac{d_2 m_s}{d_1 + d_2} & 0 & m_{u1} & 0 \\ 0 & \frac{d_1 m_s}{d_1 + d_2} & 0 & m_{u2} \end{bmatrix} \tag{19}$$

Since the mass and stiffness matrices of the VBI system include a time function, the VBI system is non-linear. This equation is solved by the Newton-Raphson scheme to simulate the dynamic responses of the vehicle and bridge models in this study.

## 2.2. Parameter Setting for Numerical Simulation

The parameters of the vehicle and bridge are shown in Tables 1 and 2. The vehicle model adopted in this numerical simulation is assumed to be a 10-ton truck. The vehicle parameters are adjusted to have a vehicle mass of 10 tons so as to maintain the dynamic characteristics of the vehicle models used in the previous studies [29,33,34]. The vehicle runs over the bridge at a constant speed of 10 m/s. The bridge is a 30 m span beam-like structure. The bridge parameters are based on a rough design process shown in the existing studies [35]. The old design procedure is adopted because old bridges are expected targets of the PRE method. It is noted that these values are decided according to the existing studies but not based on actual data. In order to avoid adopting parameters that may affect generality, such as the center of gravity being in the geometric center of the vehicle body, all parameters are also changed slightly within the range that does not significantly change their dynamic characteristics.

**Table 1.** The vehicle parameters.

| Parts | Mechanical Parameters | Symbol | Value Front ($i = 1$) | Rear ($i = 2$) | Unit |
|-------|----------------------|--------|------------|------------|------|
| Vehicle body | Mass | $m_s$ | 8310 | | kg |
| | Distance from G | $d_i$ | 1.215 | 3.185 | m |
| Suspensions | Stiffness | $k_{si}$ | 456,000 | 410,000 | kg/s$^2$ |
| | Damping | $c_{si}$ | 24,200 | 29,000 | kg/s |
| Wheels | Unsprung-mass | $m_{ui}$ | 469.0 | 751.0 | kg |
| | Stiffness | $k_{ui}$ | 4,790,000 | 4,310,000 | kg/s$^2$ |
| (General) | Speed | $v$ | 10.0 | | m/s |

**Table 2.** The bridge parameters.

| Mechanical Parameters | Symbol | Value | Unit |
|---|---|---|---|
| Span length | $L$ | 30 | m |
| Standard flexural rigidity | $\overline{EI}$ | $1.560 \times 10^{10}$ | Nm |
| Mass per unit length | $\rho A$ | 4400 | kg/m |
| Rayleigh damp. coef. | $\alpha$ | 0.7024 | |
| | $\beta$ | 0.0052 | |

In Scenarios 2 to 12, the bridge model is simulated by globally or partially reducing the flexural rigidity $EI(x)$, as shown in Table 3. In this numerical simulation, twelve patterns are prepared to consist of 1 model without stiffness reduction and 11 models with stiffness reduction cases. The bold font in this table indicates stiffness reduction.

**Table 3.** The bridge's stiffness distribution patterns.

| Scenario | | 1 | 2 | 3 | 4 | 5 | 6 | 7 | 8 | 9 | 10 | 11 | 12 |
|---|---|---|---|---|---|---|---|---|---|---|---|---|---|
| Stiffness Reduction Range: | | | (0~L) | | | (3L/7~4L/7) | | | (4L/7~6L/7) | | | (6L/7~7L/7) | |
| | $EI_1/\overline{EI}$ | 1.0 | **0.9** | **0.8** | 1.0 | 1.0 | 1.0 | 1.0 | 1.0 | 1.0 | 1.0 | 1.0 | 1.0 |
| | $EI_2/\overline{EI}$ | 1.0 | **0.9** | **0.8** | 1.0 | 1.0 | 1.0 | 1.0 | 1.0 | 1.0 | 1.0 | 1.0 | 1.0 |
| | $EI_3/\overline{EI}$ | 1.0 | **0.9** | **0.8** | 1.0 | 1.0 | 1.0 | 1.0 | 1.0 | 1.0 | 1.0 | 1.0 | 1.0 |
| Flexural rigidity: | $EI_4/\overline{EI}$ | 1.0 | **0.9** | **0.8** | **0.9** | **0.7** | **0.4** | 1.0 | 1.0 | 1.0 | 1.0 | 1.0 | 1.0 |
| | $EI_5/\overline{EI}$ | 1.0 | **0.9** | **0.8** | 1.0 | 1.0 | 1.0 | **0.9** | **0.7** | **0.4** | 1.0 | 1.0 | 1.0 |
| | $EI_6/\overline{EI}$ | 1.0 | **0.9** | **0.8** | 1.0 | 1.0 | 1.0 | **0.9** | **0.7** | **0.4** | 1.0 | 1.0 | 1.0 |
| | $EI_7/\overline{EI}$ | 1.0 | **0.9** | **0.8** | 1.0 | 1.0 | 1.0 | 1.0 | 1.0 | 1.0 | **0.9** | **0.7** | **0.4** |

Scenario 1 is provided as the standard bridge model with spatially constant stiffness. Scenario 2 and Scenario 3 indicate global stiffness reduction of the beam. The stiffness reduction ratios due to the global structural changes are set at 10% and 20%, respectively. The assumed abnormality of global stiffness reduction can include corrosion on bridges near the coast. Scenario 4, Scenario 5, and Scenario 6 provide the bridge models with different local stiffness reductions at the midspan. These scenarios can assume bending cracks on RC bridges. This kind of damage may develop for long periods without detection. Thus, the stiffness reductions in these scenarios are set at 10%, 30%, and 60%, respectively. Scenario 7, Scenario 8, and Scenario 9 indicate partial area stiffness reduction. These scenarios include the deterioration of concrete slabs. Since horizontal cracks in slabs cannot be visually detected, the vibration-based SHM is strongly expected to be used for detecting slab cracks. Scenario 10, Scenario 11, and Scenario 12 indicate local stiffness reduction at the edge. It is well known that the most common abnormality in steel girders is edge fatigue cracks.

It should be noted that these scenarios with different stiffness reductions are provided just for a parametric study and are determined independently from any actual bridge structural abnormality.

The used road unevenness $R(x)$ is shown in Figure 3. This road unevenness is generated by Monte Carlo simulation based on the actual road profile characteristics shown in the previous study [29].

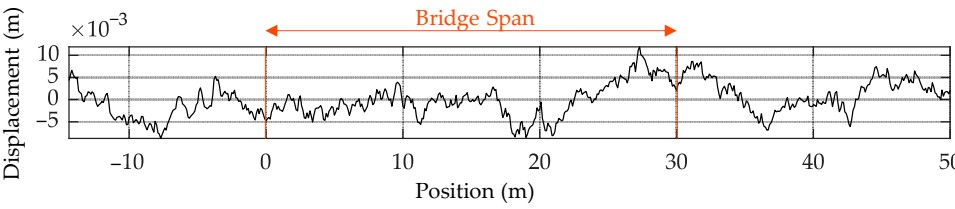

**Figure 3.** The road unevenness $R(x)$ used in the numerical simulation.

The parameters of the numerical simulation are shown in Table 4. The dynamic simulation is performed by the Newmark-beta method [36]. The time increment $\Delta t$ is 0.001. Since VBI system is generally non-linear and convergence calculation is needed, the iterative tolerance is set at 0.001. The adopted finite element model of the bridge consists of 7 beam elements. The algorithm of PRE simulation is shown in Figure 4. In the numerical simulation, the vehicle starts running from $x_1(0) = -10$ m ($x_2(0) = -14.4$ m), and the road profiles generate vehicle vibrations. When the front wheel enters the bridge ($x_1 = 0$), the interaction between the vehicle and the bridge starts working. The interaction ends when the rear wheel reaches the end of the bridge span ($x_2 = L$). The vehicle vibration observed while passing through the bridge includes information about the bridge.

**Table 4.** The simulation parameters.

| Newmark-β Method | | Time Increment | Convergence Tolerance | Bridge Element Number |
|:---:|:---:|:---:|:---:|:---:|
| $\gamma$ | $\beta$ | | | |
| 1/2 | 1/4 | $\Delta t = 0.001$ | $10^{-3}$ | 7 |

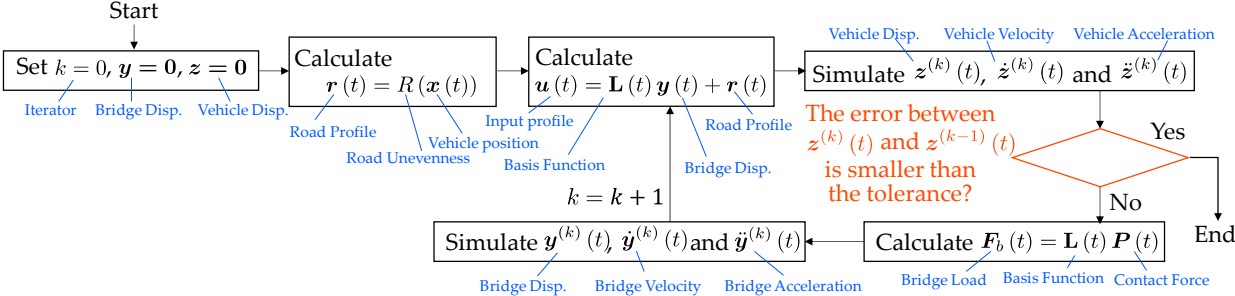

**Figure 4.** The algorithm of PRE simulation.

### 2.3. Simulated Dynamic Responses

The simulated vehicle vibrations and the corresponding road, bridge, and input profiles in Scenario 1 are shown in Figure 5. The blue/green and orange/brown curves represent waves of front and rear wheels. Figure 5a shows $r_i(t)$: the road profiles at the front and rear wheels. They are obtained by converting the road unevenness $R(x)$ shown in Figure 3 by using $x_i = x_i(0) + vt$. Figure 5b also shows $\widetilde{y}_i(t)$: the bridge profile of each wheel. The bridge profiles are the bridge deflection components $y(x_i(t), t)$ that can be calculated by Equation (16). Figure 5c shows $u_i(t)$: the input profiles. These input profiles generate dynamic vehicle responses. The generated vehicle vibration displacements, velocities, and accelerations are shown in Figure 5d–f, respectively.

According to Figure 5a–c, the bridge profiles $\widetilde{y}_i(t)$ are much smaller than road profiles $r_i(t)$. Since the road profiles $r_i(t)$ are predominant in the input profiles $u_i(t)$, road profiles $r_i(t)$ predominantly affect the vehicle responses. The displacements of the unsprung-mass $z_{ui}(t)$ shown in Figure 5d are similar to the input profiles $u_i(t)$. Furthermore, the tendency of sprung-mass responses $z_{si}(t)$ and unsprung-mass responses $z_{ui}(t)$ are similar.

In this numerical simulation, $\ddot{z}_{si}(t)$ and $\ddot{z}_{ui}(t)$ are assumed to be the measured data. $\ddot{z}_{ui}(t)$ is much larger than $\ddot{z}_{si}(t)$, but the same tendency is often observed in actual vehicles. While the unsprung responses are much larger than 1 G (=9.8 m/s$^2$), the suspensions absorb strong vibrations and play an essential role in protecting passengers and cargo.

Figure 6 also shows the vehicle vibrations in each case. There are four curves in each figure, but the differences are tiny. According to these figures, it is not easy to recognize the differences between the prepared scenarios. In SHM, low sensitivity of vibration to local damages like these results is a prevalent technical issue.

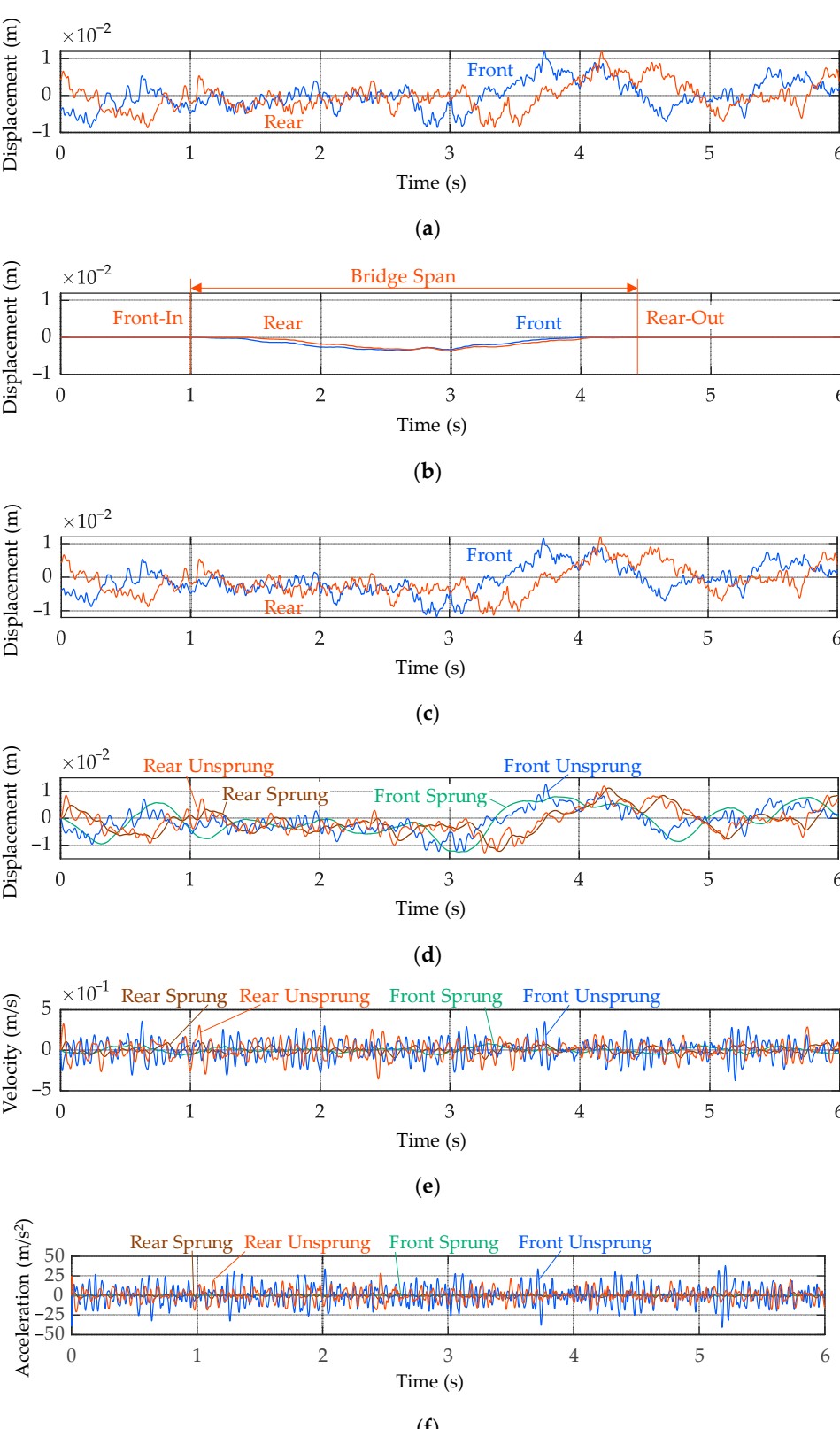

**Figure 5.** The vehicle vibration accelerations, velocities and displacements, and profiles: (**a**) road profiles $r(t) = \{r_i(t)\} = \{R(x_i(t))\}$; (**b**) bridge profiles $\widetilde{y}(t) = \{y(x_i(t), t)\} = \{N(x_i(t)) \cdot y(t)\}$; (**c**) input profiles $u(t)$; (**d**) vehicle vibration displacement $z(t)$; (**e**) vehicle vibration velocity: $\dot{z}(t)$; (**f**) vehicle vibration acceleration: $\ddot{z}(t)$.

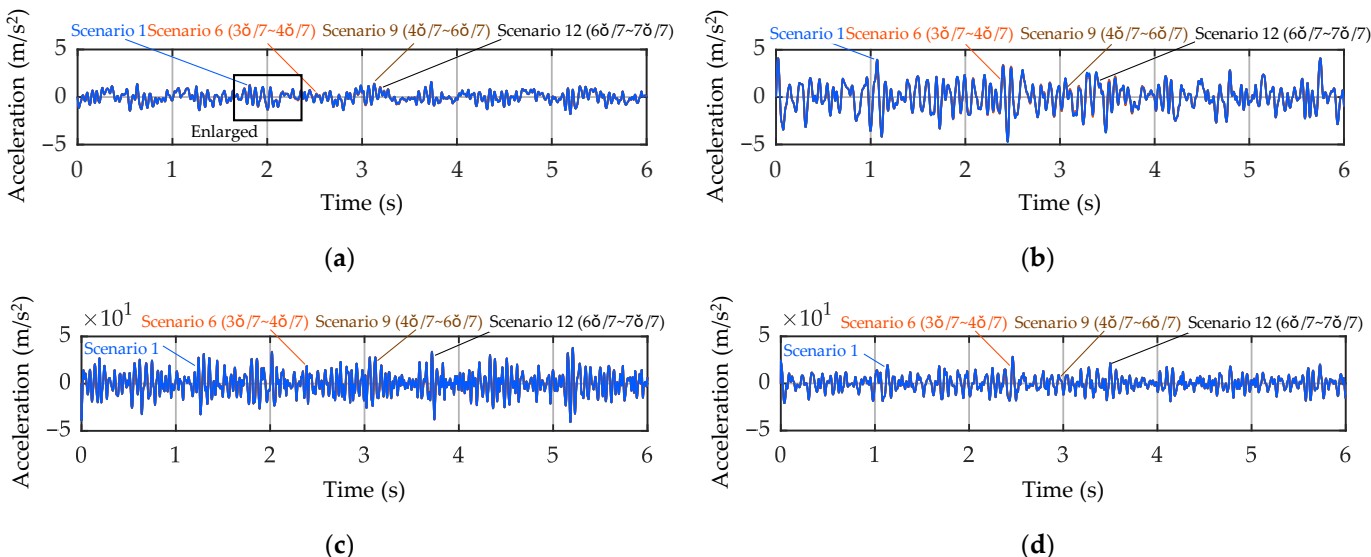

**Figure 6.** The vehicle vibration accelerations for each case: Scenario 1, Scenario 6, Scenario 9, and Scenario 12: (**a**): $\ddot{z}_{s1}$; (**b**) $\ddot{z}_{s2}$; (**c**) $\ddot{z}_{u1}$; and (**d**) $\ddot{z}_{u2}$.

Enlarged figures of the part of the simulated waveforms are provided in Appendix A for readers' easy recognition. The curves shown in Figure 5e,f are separately shown in Figures A1 and A2, respectively. The boxed part of Figure 6a is also shown in Figure A3.

The natural frequencies of the vehicle and the bridge are shown in Table 5. These natural frequencies are obtained by convergence calculation. According to this table, the natural frequencies of the bridge change after stiffness reduction.

**Table 5.** The natural frequencies of the vehicle and bridge (Unit: Hz).

| Mode Order | Vehicle | Bridge | | | | | | | | | | | |
|---|---|---|---|---|---|---|---|---|---|---|---|---|---|
| | | 0 | | | (3L/7~4L/7) | | | (4L/7~6L/7) | | | (6L/7~7L/7) | | |
| | | 1 | 2 | 3 | 4 | 5 | 6 | 7 | 8 | 9 | 10 | 11 | 12 |
| 1 | 1.12 | 3.37 | 3.20 | 3.01 | 3.32 | 3.18 | 2.83 | 3.31 | 3.15 | 2.73 | 3.37 | 3.36 | 3.25 |
| 2 | 2.04 | 14.49 | 13.75 | 12.96 | 14.48 | 14.42 | 14.25 | 14.18 | 13.46 | 12.07 | 14.44 | 14.29 | 12.84 |
| 3 | 12.63 | 36.30 | 34.44 | 32.47 | 35.86 | 34.82 | 32.58 | 35.79 | 34.49 | 31.12 | 36.05 | 35.39 | 30.69 |
| 4 | 16.84 | 71.32 | 67.66 | 63.79 | 70.84 | 69.47 | 65.17 | 70.21 | 67.45 | 61.06 | 70.77 | 69.34 | 62.06 |

However, it is still quite difficult to distinguish the changes due to stiffness reductions in the frequency domain, as shown in Figure 7. This is one of the reasons why system identification is required in drive-by bridge monitoring. From Table 5 and Figure 7, the vehicle's third and fourth natural frequencies are close to the second natural frequency of the bridge. Thus, the dynamic response components of the vehicle around the bridge's second natural frequency from 12 Hz to 15 Hz tend to be predominant.

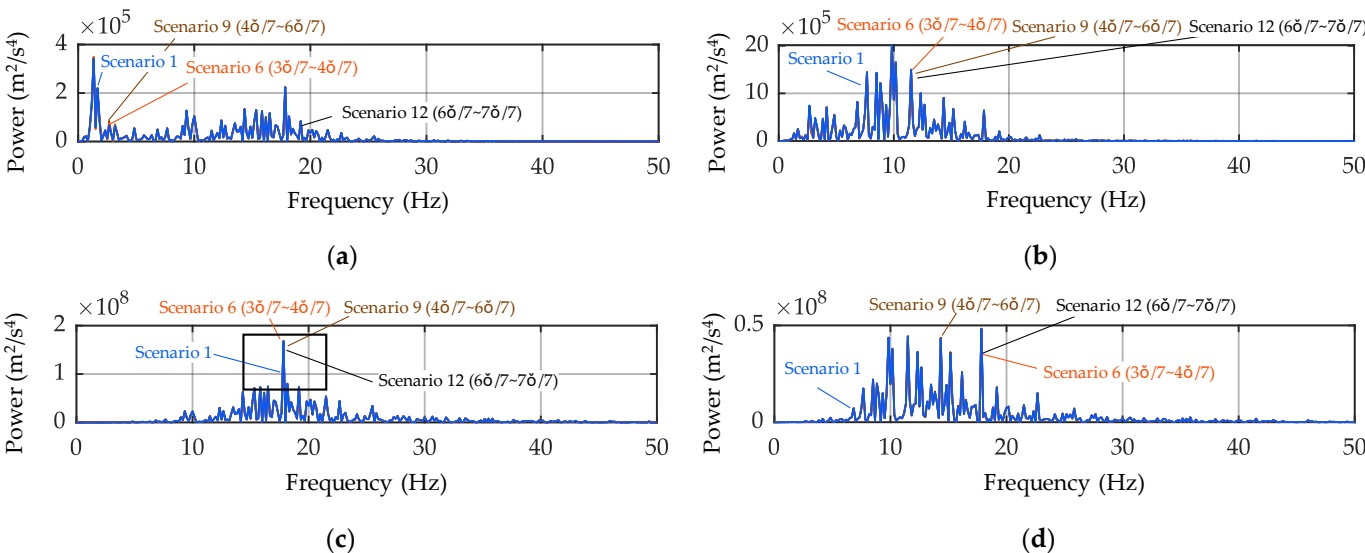

**Figure 7.** The Fourier's power spectra of the vehicle vibration accelerations for each case: Scenario 1, Scenario 6, Scenario 9, and Scenario 12: (**a**) $\ddot{z}_{s1}$; (**b**) $\ddot{z}_{s2}$; (**c**) $\ddot{z}_{u1}$; (**d**) $\ddot{z}_{u2}$. The differences between the obtained vibrations of the focused scenarios are tiny, even in the frequency domain.

## 3. The Proposed Method

### 3.1. Algorithm of the Proposed Method

The PRE method consists of optimization and numerical simulation processes. First, the obtained vehicle vibration data and randomly assumed system parameters are substituted simultaneously into Equations (1) and (13). Since all terms of the left-hand side of Equation (1), which represents the equations of motion of the vehicle, become known by this operation, the right-hand side can be calculated to estimate input profiles $u(t)$. Thus, this process can be called the IEP (input estimation problem) of the vehicle. At the same time, Equation (12), which represents the equations of motion of the bridge, can be solved because the external forces generated by the vehicle accelerations are given. This process can be called DRS (dynamic response simulation) of the bridge. By solving these two processes numerically, the road profiles $r(t)$ can be estimated:

$$\hat{R}_i(x_i(t)) = \hat{r}_i(t) \tag{20}$$

where (ˆ) denotes the estimated value. This study assumes that $\hat{R}_1(x)$: the estimated unevenness from the front wheel and $\hat{R}_2(x)$: the estimated unevenness from the rear wheel match. However, it cannot be expected that the estimated road unevenness $R_1(x)$ and $R_2(x)$ match from the beginning of the processes because the system parameters are randomly assumed. Thus, the objective function to be minimized in the PRE method can be defined as the following formula:

$$J = \int_{x_0}^{x_1} \{R_1(x) - R_2(x)\}^2 \tag{21}$$

The PRE method is a method to minimize this road unevenness error by repeating random-assuming of the mechanical parameters. The algorithm is shown in Figure 8.

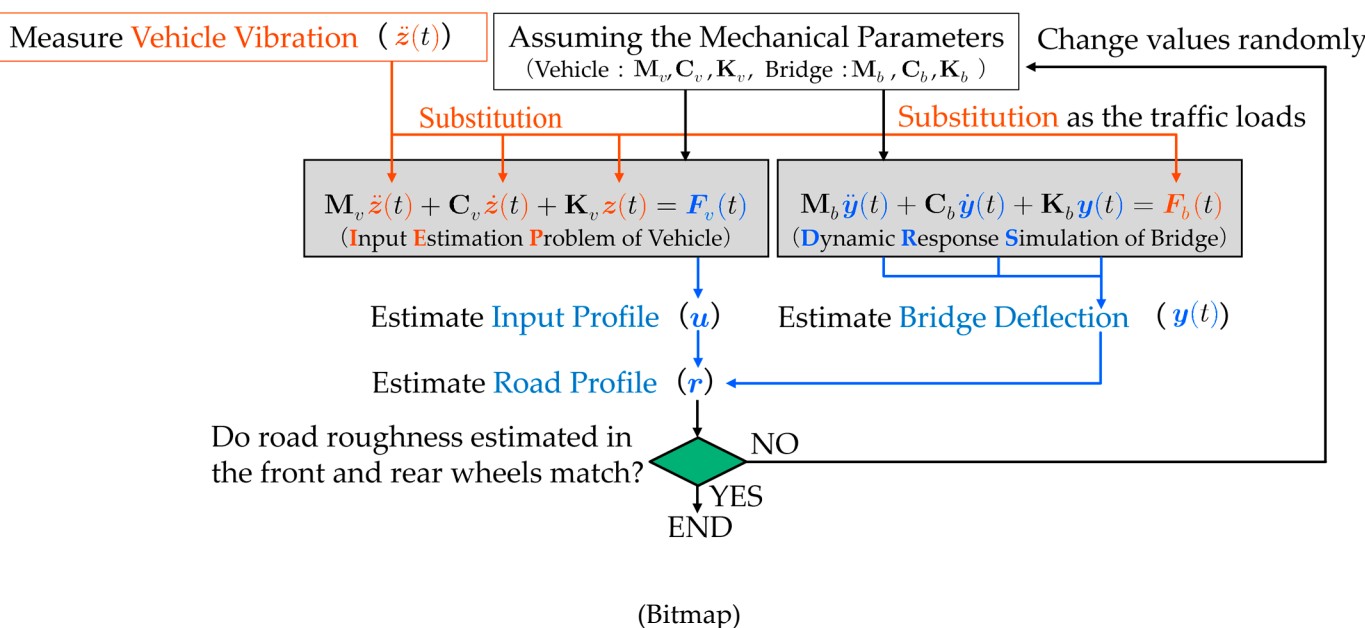

**Figure 8.** The algorithm of the PRE method.

Note that the random assumption process can use prior information about the mechanical parameters. If there is no information, a subjective probability distribution can be given because the updating process can transform the prior information into a more accurate posterior probability distribution; the randomly assumed variables converge to more likely values. However, this updating process requires a function to evaluate the likelihood of the assumed parameters. In this simulation, the uniform probability distribution is adapted to the prior distributions of the mechanical parameters. The previous studies that succeed in parameter and unevenness estimation [27,29–31] have already justified the process of random assumption.

In this study, particles are randomly generated in the 19-dimensional space. Each axis indicates each mechanical parameter: $d_1$, $c_{s1}$, $c_{s2}$, $k_{s1}$, $k_{s2}$, $m_{u1}$, $m_{u2}$, $k_{u1}$, $k_{u2}$, $EI_1$, $EI_2$, $EI_3$, $EI_4$, $EI_5$, $EI_6$, $EI_7$, $\rho A$, $\alpha$, and $\beta$. Only two parameters $M = m_s + m_{u1} + m_{u2}$, and $D = d_1 + d_2$ are assumed to be known because this process needs one known parameter at least. The total mass (weight) $M$ and the wheelbase $D$ can be easily measured. The coordinate of the considered 19-dimensional space can be expressed as

$$\boldsymbol{X} = [d_1,\ c_{s1},\ c_{s2},\ k_{s1},\ k_{s2},\ m_{u1},\ m_{u2},\ k_{u1},\ k_{u2},\ EI_1,\ EI_2,\ EI_3,\ EI_4,\ EI_5,\ EI_6,\ EI_7,\ \rho A, \alpha,\ \beta] \tag{22}$$

Let $\boldsymbol{X}_i^{(k)}$ be the coordinate of the *i*-th particle in the *k*-th step. $\boldsymbol{X}_i^{(0)}$ is generated as a uniform random number ranging from 0.8 to 1.2 times the correct values. This means that the uncertainty of the prior information of each parameter is assumed to be 20%. If more accurate prior information is available, the PRE method can use it. Keenahan et al. [27] apply the Gaussian distributions in the previous study. However, this paper assumes uniform distributions to confirm the PRE's capability because the uniform distribution can be an unfavorable condition for the updating scheme.

The coordinates of each particle are updated according to the following formula:

$$\boldsymbol{X}_i^{(k+1)} = \begin{cases} \boldsymbol{X}_i^{(k)} + \Delta \boldsymbol{X}_i^{(k)} & \text{if } J\left(\boldsymbol{X}_i^{(k)} + \Delta \boldsymbol{X}_i^{(k)}\right) \le J\left(\boldsymbol{X}_i^{(k)}\right) \\ \boldsymbol{X}_i^{(k)} & \text{if } J\left(\boldsymbol{X}_i^{(k)} + \Delta \boldsymbol{X}_i^{(k)}\right) > J\left(\boldsymbol{X}_i^{(k)}\right) \end{cases} \tag{23}$$

where $\Delta \boldsymbol{X}_i^{(k)}$ is random. $J(\boldsymbol{X})$ is the road unevenness residual calculated from $\boldsymbol{X}$. This process is a kind of the MCMC (Monte Carlo Markov chain) method called the random-walk algorithm.

### 3.2. Accuracy Evaluation

The MCMC-powered PRE method gives the estimation results of the mechanical parameters as probability density distributions. Therefore, it is necessary to invent a new index for accuracy evaluation. Suppose a normalized estimate of a mechanical parameter is given as $a$. $E[a]$ denotes the average of the estimated value $a$. If its correct value is $a_0$, the error can be evaluated by $\varepsilon = \sqrt{(E[a] - a_o)^2}$. Thus, the accuracy evaluation index $S_A$ can be given by the following formula:

$$S_A = 1 - \sqrt{(E[a] - a_o)^2} \tag{24}$$

### 3.3. Application of the Proposed Method

Table 6 shows the simulation parameters of the PRE process. In the vehicle's IEP, the numerical integration of $\ddot{z}(t)$ to calculate $\dot{z}(t)$ and $z(t)$ is needed. This numerical integration is done by the Newmark-β method. On the other hand, the DRS of the bridge also adopts the Newmark-β method to simulate the bridge's dynamic responses from the vehicle vibrations. To search for the optimized solution, the MCMC method, of which the number of Particles $X_i^{(k)}$ is 500, and of which the maximum step number is 600, is applied. The number of particles for the MCMC process depends on the computing environment. Since almost no updates are observed after 600 steps, the end step is decided, as shown in Table 6.

**Table 6.** The parameters of the PRE method with MCMC simulation.

| Process | Parameter | Value |
|---|---|---|
| Dynamic simulation and numerical integration | The Newmark-β method | $\gamma = 1/2$ $\beta = 1/4$ |
| | Time increment | $\Delta t = 0.001$ |
| Monte Carlo Markov chain method | Particle number | 500 |
| | Step number | 600 |

## 4. Results and Discussions

Figure 9 shows the application results of the PRE method in Scenario 1 (no stiffness reduction). It consists of the figures indicating the probability density distributions of the estimated mechanical parameters of the VBI system. Each vertical axis indicates the probability density of each estimated parameter, while each horizontal axis denotes the normalized value. The red and blue bars in each histogram represent the prior and posterior distributions, respectively. The posterior distributions are obtained by updating the particles $X_i^{(k)}$ with the MCMC-powered PRE method. The left and right show the vehicle's and the bridge's mechanical parameters. Noted that the coordinate $X_i^{(k)}$ does not include the vehicular body mass: $m_{s1}$ and $m_{s2}$ but $d_1$. Since $M$ and $D$ are fixed, $m_{s1}$ and $m_{s2}$ are automatically decided when $d_1$ is given. Because of the distance from G: $d_1$ is randomly assumed to range from 0.8 to 1.2 times the correct value, the normalized prior probabilities of $m_{si}$ in the initial step do not distribute from 0.8 to 1.2.

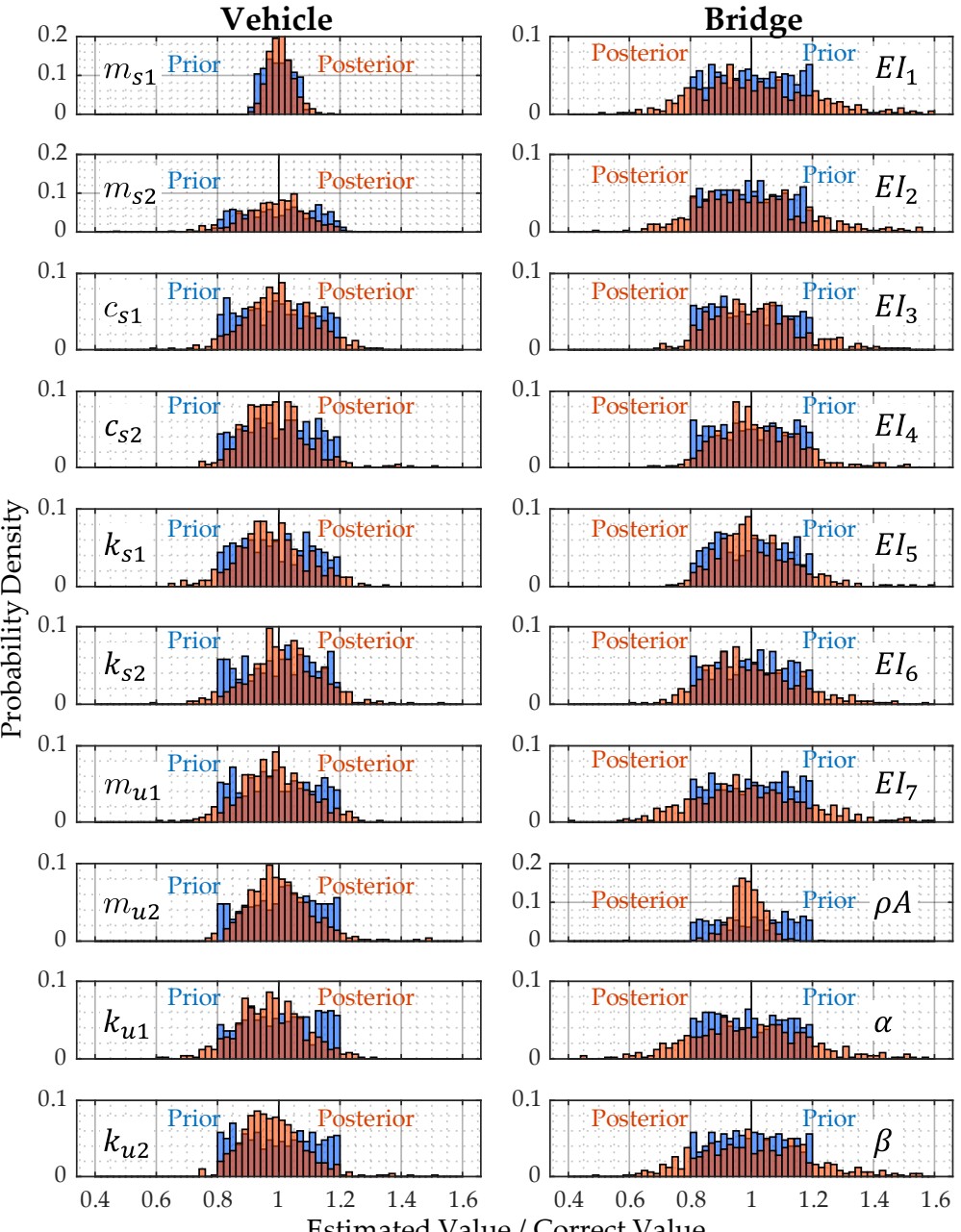

**Figure 9.** The application results of the PRE method in Scenario 1: these figures indicate the prior and posterior probability density distributions of the estimated mechanical parameters. The blue bars represent the prior information assuming uniform probability distributions. The red bars represent the posterior information updated by the MCMC process. The left and right indicate the vehicle and bridge mechanical parameters, respectively.

According to Figure 9, the updated distributions peak around 1.0, which means that the PRE method can estimate these mechanical parameters. The posterior probability density distributions of all parameters except bridge damping coefficients $\alpha$ and $\beta$ are improved from the prior information by the MCMC process.

Figure 10 also shows the correct and estimated road unevenness $R(x)$, $\hat{R}_1(x)$ and $\hat{R}_2(x)$ by the PRE method using the optimal particle $X_i^{(600)}$ of which $J$ value is minimum. In this study, it is confirmed that when substituting correct values instead of assumed values, $J$ becomes zero. If the updated values are close to the correct values in the PRE process, the difference between the estimated road unevenness functions must be almost zero.

However, there is no guarantee that the matching two estimated unevenness functions are also close to the correct waveform. However, according to Keenahan et al. [27], the Monte Carlo-based process reproduces the correct road unevenness, while Figure 10 also shows the high estimation accuracy of road unevenness $R(x)$. This means that the same tendency in the previous study [27] is confirmed; when the estimated road unevenness functions match, both also match the correct one. Their tiny errors and differences are shown in Figure A4, Appendix A.

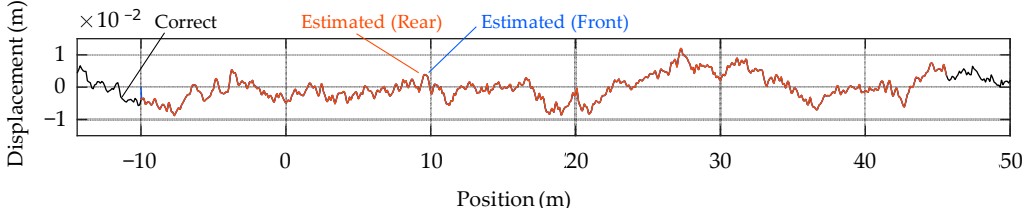

**Figure 10.** The estimated road unevenness $R(x)$: The estimated road unevenness $\hat{R}_1(x)$ and $\hat{R}_2(x)$ shown in this figure are calculated using the optimal particle $\boldsymbol{X}_{best}^{(600)}$ minimizing $J$ in Scenario 1.

Figures 11 and 12 are the results obtained from the case of Scenario 6. In this scenario, $EI_4$ is reduced in numerical simulation. According to these figures, the PRE method can estimate the stiffness reduction of the stiffness decrease element. This result also suggests that the PRE method can update the parameters to the correct values if wrong prior information is given. However, the estimated stiffness $EI_4$ distributes higher than the correct value of 0.6 in this scenario. This overestimation should be improved in future works because it can lead to a severe accident. On the other hand, according to Figure 11, the estimated stiffness $EI_3$ and $EI_5$ are underestimated. While this underestimation means that the estimation accuracy becomes lower than that of Scenario 1 (no stiffness reduction), it can be taken positively because it increases the possibility of reduced stiffness detection. These results imply that the PRE method partially succeeds in stiffness reduction estimation.

According to Figure 12, the road unevenness can also be estimated accurately in Scenario 6. This means that the estimation accuracy of the PRE method does not change when the bridge's midspan stiffness decreases. The errors and residuals between the correct and estimated road unevenness functions are also shown in Figure A4, Appendix A.

Next, Figure 13 shows the probability density distributions of all estimated mechanical parameters in (a) Scenario 1, (b) Scenario 2, and (c) Scenario 3, respectively. Figure 13a reprints Figure 9. Note that the bridge stiffness $EI_i$ is divided by the standard value $\overline{EI}$. Thus, when the PRE method accurately estimated the reduced stiffness of the bridge, the normalized values of $EI_i$ in Scenario 2 and Scenario 3 should equal 0.9 and 0.8. In Figure 13b,c, the bridge stuffiness is assumed to be reduced globally by 10% and 20%. It can be confirmed that the estimated values of 20% less stiffness of Scenario 3 tend to be distributed lower than those of 10% less stiffness of Scenario 2. This result implies the feasibility of the PRE method not only to detect but also to estimate the stiffness reductions of bridges. However, $EI_1$ and $EI_7$ are not estimated accurately. The PRE method cannot work well where the amplitude of bridge vibration is small.

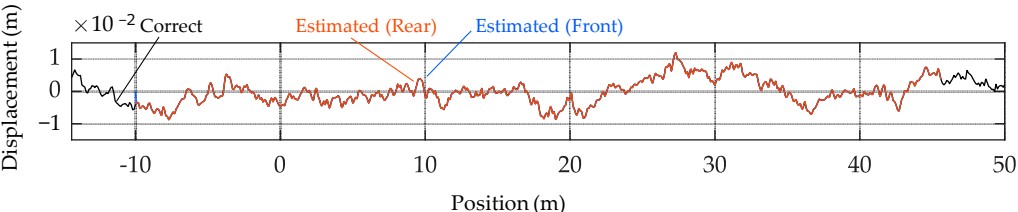

**Figure 11.** The application results of the PRE method in the case of Scenario 6 (stiffness decrease section: $3L/7\sim4L/7$, decrease ratio: 60%): these figures indicate the prior and posterior probability density distributions of the estimated mechanical parameters. The MCMC process can react to the reduction of $EI_4$, while the posterior distributions of $EI_3$ and $EI_5$ also shift to lower.

**Figure 12.** The estimated road unevenness $R(x)$: the estimated road unevenness $\hat{R}_1(x)$ and $\hat{R}_2(x)$ shown in this figure are calculated using the optimal particle $X_{best}^{(600)}$ minimizing $J$ in Scenario 6.

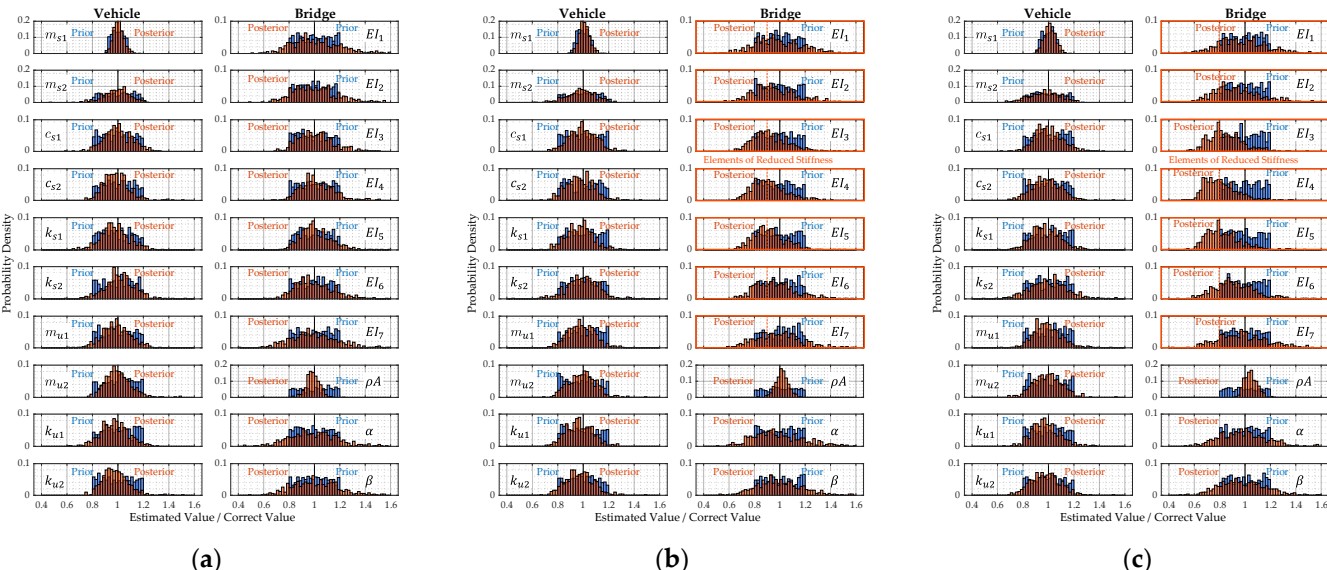

**Figure 13.** The application results of the PRE method: (**a**) Scenario 1; (**b**) Scenario 2 (stiffness decrease section: 0~*L*, decrease ratio: 10%); and (**c**) Scenario 3 (stiffness decrease section: 0~*L*, decrease ratio: 20%).

The bridge's density $\rho A$ can be estimated accurately. However, this study assumes that the density $\rho A$ is constant regardless of location $x$. If the density $\rho A$ is divided into elements as well as the stiffness $EI_i$, the estimation accuracy of the density $\rho A$ and the relationship between the density and stiffness estimation accuracy should be examined in future work. The damping coefficients $\alpha$ and $\beta$ cannot be estimated in the shown scenarios. It is believed that these values do not significantly affect the dynamic behavior of the VBI system.

According to Figure 13, the vehicle parameters in each scenario can also be estimated accurately. Because the estimation accuracy of bridge vibration decreases, the accuracy of vehicle parameters could be affected. However, the vehicle estimation accuracy in these scenarios does not change after the bridge's stiffness reductions. It can be said that the estimation accuracy of the vehicle parameters is not affected by the bridge's state.

Figure 14 shows the probability density distributions of all estimated mechanical parameters in (a) Scenario 4, (b) Scenario 5, and (c) Scenario 6, respectively. According to these figures, the stiffness reduction at midspan can be detected. Figure 14c reprints Figure 11. In these scenarios, the applicability of the PRE method to the estimation of local stiffness reduction of the bridge.

It is not easy to recognize the 10% stiffness reduction in Figure 14a, while the 30% and 60% decreases can be observed in Figure 14b,c. Estimating the slight ratio of the local stiffness reduction requires the quantitative evaluation of the results, while the global reduction can be observed in Figure 13b.

Figure 15 shows the probability density distributions of all estimated mechanical parameters in (a) Scenario 7, (b) Scenario 8, and (c) Scenario 9, respectively. These cases indicate the ranging stiffness reductions and provide a discussion basis for the PRE applicability to the width of the stiffness reduction. The PRE method can recognize two elements with reduced stiffness from these sub-figures.

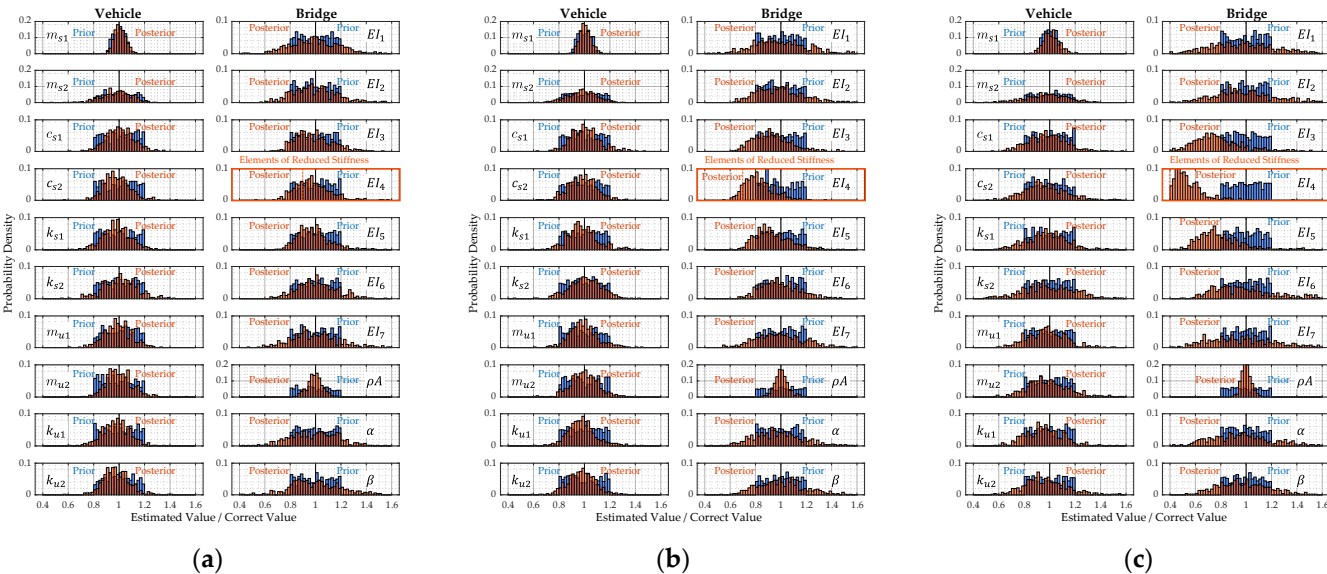

**Figure 14.** The application results of the PRE method: (**a**) Scenario 4 (stiffness decrease section: $3L/7{\sim}4L/7$, decrease ratio: 10%); (**b**) Scenario 5 (stiffness decrease section: $3L/7{\sim}4L/7$, decrease ratio: 30%); and (**c**) Scenario 6 (stiffness decrease section: $3L/7{\sim}4L/7$, decrease ratio: 60%).

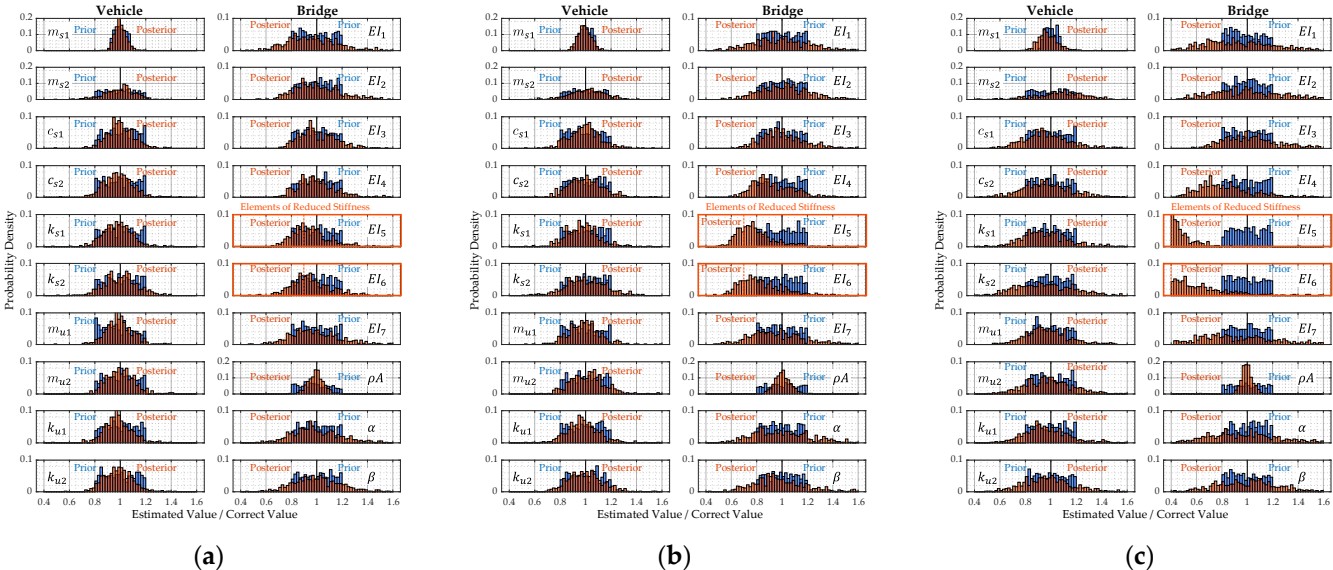

**Figure 15.** The application results of the PRE method: (**a**) Scenario 7 (stiffness decrease section: $4L/7{\sim}6L/7$, decrease ratio: 10%); (**b**) Scenario 8 (stiffness decrease section: $4L/7{\sim}6L/7$, decrease ratio: 30%); and (**c**) Scenario 9 (stiffness decrease: $4L/7{\sim}6L/7$, decrease ratio: 60%).

Figure 16 shows the probability density distributions of all estimated mechanical parameters in (a) Scenario 10, (b) Scenario, 11 and (c) Scenario 12, respectively. These cases indicate edge stiffness reduction. According to these figures, the tendency of stiffness reduction cannot be detected from the estimated distributions of $EI_7$. However, the next elements have underestimated values. This underestimation can help detect stiffness reduction at the bridge's edge.

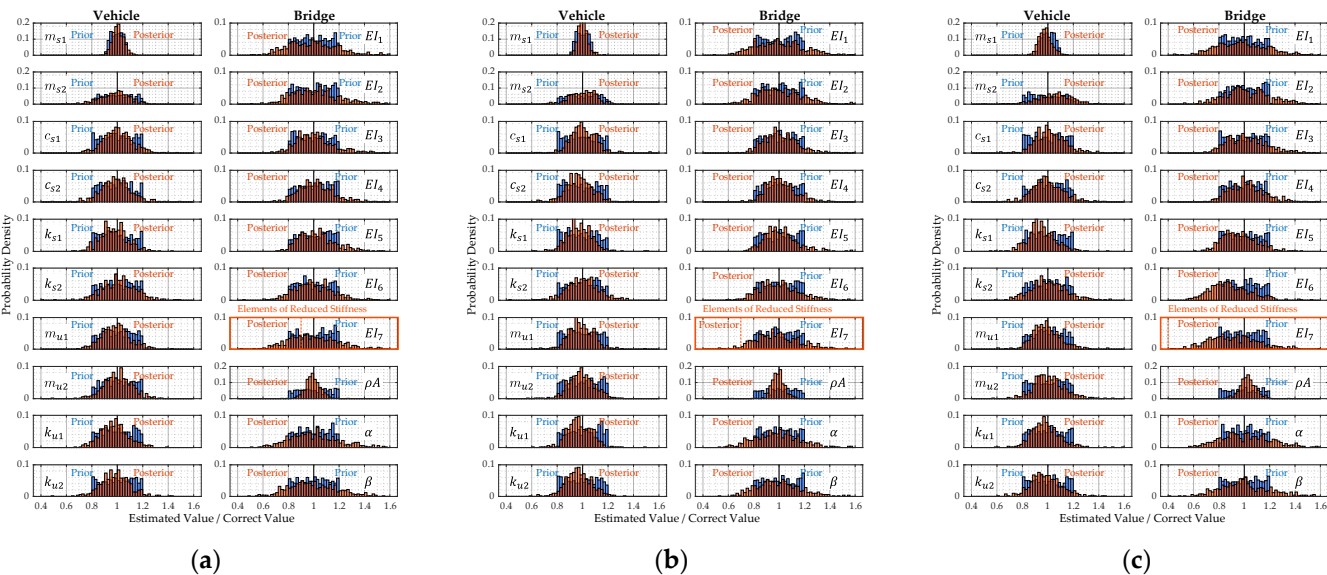

(**a**)                       (**b**)                       (**c**)

**Figure 16.** The application results of the PRE method: (**a**) Scenario 10 (stiffness decrease section: $4L/7 \sim 6L/7$, decrease ratio: 10%); (**b**) Scenario 11 (stiffness decrease section: $4L/7 \sim 6L/7$, decrease ratio: 30%); (**c**) Scenario 12 (stiffness decrease section: $4L/7 \sim 6L/7$, decrease ratio: 60%).

Table 7 shows the average and standard deviation of the estimated normalized mechanical parameters. The bold style in this table indicates the average of the estimated stiffness at the elements with reduced stiffness. The obtained average can be used to evaluate the estimation accuracy. On the other hand, the standard deviation indicates the reliability of the obtained distribution. According to the averages in this table, the reduced stiffness of the bridge can be directly detected except in the edge elements. It is also noted that the values of the reduced stiffness are often overestimated.

**Table 7.** The average and standard deviation of the estimated parameters: When the average of estimated $EI_i$ is shown in bold style, this indicates that the element $i$ has reduced stiffness.

| | Scenario 1 | | Scenario 2 | | Scenario 3 | | Scenario 4 | | Scenario 5 | | Scenario 6 | | Scenario 7 | | Scenario 8 | | Scenario 9 | | Scenario 10 | | Scenario 11 | | Scenario 12 | |
|---|---|---|---|---|---|---|---|---|---|---|---|---|---|---|---|---|---|---|---|---|---|---|---|---|
| | ave. | std. | ave. | std. | ave. | std. | ave. | std. | ave. | std. | ave. | std. | ave. | std. | ave. | std. | ave. | std. | ave. | std. | ave. | std. | ave. | std. |
| $m_{s1}$ | 1.05 | 0.01 | 1.07 | 0.02 | 0.98 | 0.01 | 1.05 | 0.01 | 1.08 | 0.01 | 1.06 | 0.02 | 0.99 | 0.02 | 0.94 | 0.02 | 0.90 | 0.05 | 1.01 | 0.02 | 1.02 | 0.02 | 0.97 | 0.02 |
| $m_{s2}$ | 0.86 | 0.03 | 0.83 | 0.05 | 1.05 | 0.03 | 0.87 | 0.03 | 0.78 | 0.02 | 0.86 | 0.05 | 1.03 | 0.06 | 1.16 | 0.05 | 1.27 | 0.13 | 0.97 | 0.05 | 0.94 | 0.04 | 1.09 | 0.04 |
| $c_{s1}$ | 1.00 | 0.15 | 1.00 | 0.14 | 1.00 | 0.17 | 1.01 | 0.17 | 1.01 | 0.16 | 0.98 | 0.16 | 1.00 | 0.18 | 0.99 | 0.18 | 0.98 | 0.21 | 1.00 | 0.16 | 1.01 | 0.15 | 1.03 | 0.18 |
| $c_{s2}$ | 0.99 | 0.15 | 0.97 | 0.15 | 0.98 | 0.17 | 0.98 | 0.16 | 0.98 | 0.15 | 0.97 | 0.19 | 0.99 | 0.17 | 0.99 | 0.18 | 0.97 | 0.20 | 0.98 | 0.16 | 0.97 | 0.14 | 1.01 | 0.19 |
| $k_{s1}$ | 0.98 | 0.15 | 0.98 | 0.14 | 0.98 | 0.17 | 0.99 | 0.17 | 0.99 | 0.16 | 0.98 | 0.19 | 0.99 | 0.18 | 0.97 | 0.18 | 0.96 | 0.21 | 0.97 | 0.15 | 0.97 | 0.14 | 0.96 | 0.17 |
| $k_{s2}$ | 1.01 | 0.15 | 1.00 | 0.17 | 1.00 | 0.19 | 1.01 | 0.18 | 1.01 | 0.17 | 1.00 | 0.22 | 1.00 | 0.19 | 1.01 | 0.20 | 0.96 | 0.24 | 1.01 | 0.18 | 1.00 | 0.16 | 1.01 | 0.19 |
| $m_{u1}$ | 0.99 | 0.14 | 0.99 | 0.14 | 1.00 | 0.17 | 1.00 | 0.17 | 0.99 | 0.15 | 0.97 | 0.17 | 1.00 | 0.18 | 0.98 | 0.17 | 0.98 | 0.20 | 0.99 | 0.16 | 0.98 | 0.14 | 0.98 | 0.17 |
| $m_{u2}$ | 1.01 | 0.14 | 0.99 | 0.14 | 1.00 | 0.17 | 1.00 | 0.16 | 1.00 | 0.15 | 0.99 | 0.19 | 1.00 | 0.18 | 1.01 | 0.18 | 0.98 | 0.20 | 1.01 | 0.15 | 0.99 | 0.14 | 1.01 | 0.17 |
| $k_{u1}$ | 0.98 | 0.14 | 0.98 | 0.15 | 0.98 | 0.17 | 0.97 | 0.17 | 0.99 | 0.15 | 0.96 | 0.16 | 0.98 | 0.18 | 0.98 | 0.17 | 0.96 | 0.20 | 0.98 | 0.16 | 0.97 | 0.15 | 0.98 | 0.17 |
| $k_{u2}$ | 0.99 | 0.14 | 0.97 | 0.15 | 0.98 | 0.17 | 0.98 | 0.16 | 0.97 | 0.14 | 0.98 | 0.20 | 0.98 | 0.17 | 1.00 | 0.19 | 0.97 | 0.20 | 0.99 | 0.16 | 0.97 | 0.15 | 0.98 | 0.17 |
| $EI_1$ | 1.02 | 0.23 | **0.99** | 0.22 | **0.98** | 0.24 | 1.01 | 0.23 | 1.00 | 0.24 | 0.99 | 0.29 | 0.97 | 0.25 | 0.99 | 0.26 | 0.99 | 0.39 | 1.00 | 0.24 | 1.01 | 0.22 | 0.99 | 0.25 |
| $EI_2$ | 1.00 | 0.20 | **0.97** | 0.21 | **0.94** | 0.22 | 1.00 | 0.22 | 1.00 | 0.22 | 1.01 | 0.30 | 1.01 | 0.23 | 1.03 | 0.24 | 1.06 | 0.36 | 1.02 | 0.23 | 1.01 | 0.21 | 1.03 | 0.23 |
| $EI_3$ | 1.02 | 0.18 | **0.94** | 0.19 | **0.86** | 0.18 | 0.99 | 0.21 | 0.93 | 0.18 | 0.79 | 0.22 | 1.01 | 0.22 | 1.01 | 0.22 | 1.07 | 0.30 | 1.02 | 0.21 | 1.02 | 0.18 | 1.02 | 0.20 |
| $EI_4$ | 1.02 | 0.17 | **0.92** | 0.16 | **0.81** | 0.16 | **0.97** | 0.18 | **0.83** | 0.15 | **0.54** | 0.13 | 1.00 | 0.20 | 0.92 | 0.21 | 0.84 | 0.26 | 1.02 | 0.19 | 1.01 | 0.16 | 1.03 | 0.19 |
| $EI_5$ | 1.02 | 0.19 | **0.92** | 0.16 | **0.84** | 0.18 | 0.99 | 0.19 | 0.94 | 0.17 | 0.81 | 0.20 | **0.96** | 0.24 | **0.80** | 0.19 | **0.44** | 0.13 | 1.03 | 0.20 | 1.02 | 0.18 | 0.96 | 0.19 |
| $EI_6$ | 1.01 | 0.19 | **0.98** | 0.20 | **0.92** | 0.20 | 1.00 | 0.20 | 0.99 | 0.21 | 1.00 | 0.27 | **0.96** | 0.22 | **0.84** | 0.20 | **0.60** | 0.27 | 1.00 | 0.19 | 1.01 | 0.19 | 0.89 | 0.21 |
| $EI_7$ | 0.99 | 0.22 | **1.00** | 0.22 | **0.99** | 0.23 | 1.00 | 0.24 | 1.00 | 0.22 | 0.99 | 0.31 | 0.98 | 0.24 | 0.98 | 0.26 | 0.97 | 0.40 | **0.99** | 0.23 | **1.00** | 0.22 | **0.92** | 0.24 |
| $\rho A$ | 0.99 | 0.12 | 1.02 | 0.18 | 1.05 | 0.14 | 0.98 | 0.13 | 1.02 | 0.16 | 0.99 | 0.09 | 0.98 | 0.15 | 1.01 | 0.14 | 0.99 | 0.13 | 0.99 | 0.13 | 1.00 | 0.18 | 1.03 | 0.15 |
| $\alpha$ | 1.00 | 0.22 | 1.00 | 0.23 | 0.99 | 0.22 | 1.00 | 0.28 | 0.98 | 0.22 | 0.99 | 0.28 | 1.00 | 0.23 | 1.01 | 0.28 | 1.02 | 0.36 | 1.00 | 0.22 | 1.00 | 0.23 | 1.02 | 0.25 |
| $\beta$ | 1.01 | 0.21 | 1.01 | 0.23 | 0.98 | 0.21 | 1.01 | 0.27 | 1.01 | 0.22 | 0.99 | 0.23 | 1.00 | 0.22 | 1.00 | 0.29 | 1.03 | 0.32 | 1.00 | 0.25 | 1.00 | 0.20 | 1.01 | 0.25 |

**ave.**: average, **std.**: standard deviation

In Table 7, if a standard deviation is more significant, the reliability of the corresponding estimated parameter is lower. The standard deviations of suspensions' and tires' characteristics $c_{si}$, $k_{si}$, $m_{ui}$ and $k_{ui}$ tend to be larger than those of sprung-mass. The PRE method can estimate the vehicle's body mass $m_{si}$ more accurately than the other parameters. This tendency can be attributed to the equations of motion of the vehicle. The restoring force and damping force generated by each suspension are internal. Thus, these two acting internal forces can be eliminated from Equation (1). The accuracy improvement of these values will be a technical issue of future work.

On the other hand, the standard deviations of the bridge stiffness at both edges $EI_1$ and $EI_7$ also tend to be higher. Since the bridge vibrations at both ends should be smaller than others, it becomes difficult to evaluate their influence on vehicle responses. While the reduced stiffness values tend to be overestimated, the stiffness next to the damaged element tends to be underestimated. Using this characteristic of the PRE method, there is a feasibility of detecting edge stiffness reduction. The Rayleigh damping coefficients $\alpha$ and $\beta$ are estimated with low accuracy. This means that the effectiveness of $\alpha$ and $\beta$ to the dynamic responses of the vehicle is minimal.

In Scenario 4, Scenario 5, and Scenario 6, the flexural rigidity of the stiffness decrease element $EI_4$ tends to be lower than that of Scenario 1, while $EI_3$ and $EI_5$ also decrease despite their soundness. Similarly, in the cases of Scenario 7, Scenario 8, and Scenario 9, the bending stiffness values of the stiffness decrease elements $EI_5$ and $EI_6$ can be estimated, while $EI_4$ is also underestimated. In the cases of Scenario 10, Scenario 11, and Scenario 12, it is difficult to recognize the stiffness reduction tendency from the posterior distribution of $EI_7$, while the underestimated value of $EI_6$ can be used for the detection of stiffness reduction in Scenario 12.

Table 8 shows the estimation accuracy of the PRE method. The accuracy is evaluated by Equation (23). Values closer to 1 indicate better accuracy. The bold style indicates the estimated values of stiffness decrease elements. The estimation accuracy of the stiffness decrease elements and near elements tend to be lower. This low accuracy can be used to detect bridge stiffness reduction but not to estimate it. The problem of stiffness overestimation and underestimation may be due to the limitation of the update efficiency of the MCMC scheme. Changing the optimization scheme may improve accuracy.

It should be noted that the accuracy of $EI_1$ of Scenario 1 is 0.98. It seems to be accurate, but the standard deviation of $EI_1$ shown in Table 6 is 0.23, which is bigger than others (ranging from 0.1 to 0.2). Therefore, it can be said that the estimated $EI_1$ in this scenario is less reliable than the others. The performance of the inspection scheme should be evaluated by accuracy and reliability. According to Table 7, the accuracies of $EI_3$ and $EI_4$ are 0.79 and 0.86, respectively. While their correct values are 1.0 and 0.4, the average of $EI_3$ and $EI_4$ shown in Table 6 are 0.79 and 0.54. This means that the estimation accuracies of $EI_3$ and $EI_4$ are similar, but $EI_3$ is underestimated, while $EI_4$ is overestimated.

These results of applying the PRE process are characterized by the ability to estimate all three VBI system elements: vehicles, bridges, and road surfaces. The comparison between the performance of the PRE method and existing schemes should be considered. Yang's scheme [23], based on a one-tractor-two-trailers system, can estimate the bridge mode shape with perfect accuracy. The estimation accuracy of bridge stiffness provided by the PRE method is not as high as Yang's scheme [23]. However, Yang's method cannot extract road roughness information but erase it. The vehicle's dynamic characteristics are analyzed but not identified. The PRE method assumes to be applied to inspect vehicle conditions in the future, while Yang's method is not concerned with the state of the two monitored vehicles. However, Yang's method is superior because it can be expected to have the same accuracy as a numerical simulation, even when used in an actual environment. The PRE process still contains algorithms susceptible to noise and needs improvement. This noise vulnerability of the PRE method is confirmed in a field experiment [32], while Yang's method is verified in a field experiment.

**Table 8.** The estimation accuracy of the mechanical parameters by the PRE method: When the average of estimated $EI_i$ is shown in bold style, this indicates that the element $i$ has reduced stiffness.

| | Scenario 1 | Scenario 2 | Scenario 3 | Scenario 4 | Scenario 5 | Scenario 6 | Scenario 7 | Scenario 8 | Scenario 9 | Scenario 10 | Scenario 11 | Scenario 12 |
|---|---|---|---|---|---|---|---|---|---|---|---|---|
| $m_{s1}$ | 0.95 | 0.93 | 0.98 | 0.95 | 0.92 | 0.94 | 0.99 | 0.94 | 0.90 | 0.99 | 0.98 | 0.97 |
| $m_{s2}$ | 0.86 | 0.83 | 0.95 | 0.87 | 0.78 | 0.86 | 0.97 | 0.84 | 0.73 | 0.97 | 0.94 | 0.91 |
| $c_{s1}$ | 1.00 | 1.00 | 1.00 | 0.99 | 0.99 | 0.98 | 1.00 | 0.99 | 0.98 | 1.00 | 0.99 | 0.97 |
| $c_{s2}$ | 0.99 | 0.97 | 0.98 | 0.98 | 0.98 | 0.97 | 0.99 | 0.99 | 0.97 | 0.98 | 0.97 | 0.99 |
| $k_{s1}$ | 0.98 | 0.98 | 0.98 | 0.99 | 0.99 | 0.98 | 0.99 | 0.97 | 0.96 | 0.97 | 0.97 | 0.96 |
| $k_{s2}$ | 0.99 | 1.00 | 1.00 | 0.99 | 0.99 | 1.00 | 1.00 | 0.99 | 0.95 | 0.99 | 1.00 | 0.99 |
| $m_{u1}$ | 0.99 | 0.99 | 1.00 | 1.00 | 0.99 | 0.97 | 1.00 | 0.98 | 0.98 | 0.99 | 0.98 | 0.98 |
| $m_{u2}$ | 0.99 | 0.99 | 1.00 | 1.00 | 1.00 | 0.99 | 1.00 | 0.99 | 0.98 | 0.99 | 0.99 | 0.99 |
| $k_{u1}$ | 0.98 | 0.98 | 0.98 | 0.97 | 0.99 | 0.96 | 0.98 | 0.98 | 0.96 | 0.98 | 0.97 | 0.98 |
| $k_{u2}$ | 0.99 | 0.97 | 0.98 | 0.98 | 0.97 | 0.98 | 0.98 | 1.00 | 0.97 | 0.99 | 0.97 | 0.98 |
| $EI_1$ | 0.98 | **0.91** | **0.82** | 0.99 | 1.00 | 0.99 | 0.97 | 0.99 | 0.99 | 1.00 | 0.99 | 0.99 |
| $EI_2$ | 1.00 | **0.93** | **0.86** | 1.00 | 1.00 | 0.99 | 0.99 | 0.97 | 0.94 | 0.98 | 0.99 | 0.97 |
| $EI_3$ | 0.98 | **0.96** | **0.94** | 0.99 | 0.93 | 0.79 | 0.99 | 0.99 | 0.93 | 0.98 | 0.98 | 0.98 |
| $EI_4$ | 0.98 | **0.98** | **0.99** | **0.93** | **0.87** | **0.86** | 1.00 | 0.92 | 0.84 | 0.98 | 0.99 | 0.97 |
| $EI_5$ | 0.98 | **0.98** | **0.96** | 0.99 | 0.94 | 0.81 | **0.94** | **0.90** | **0.96** | 0.97 | 0.98 | 0.96 |
| $EI_6$ | 0.99 | **0.92** | **0.88** | 1.00 | 0.99 | 1.00 | **0.94** | **0.86** | **0.80** | 1.00 | 0.99 | 0.89 |
| $EI_7$ | 0.99 | **0.90** | **0.81** | 1.00 | 1.00 | 0.99 | 0.98 | 0.98 | 0.97 | **0.91** | **0.70** | **0.48** |
| $\rho A$ | 0.99 | 0.98 | 0.95 | 0.98 | 0.98 | 0.99 | 0.98 | 0.99 | 0.99 | 0.99 | 1.00 | 0.97 |
| $\alpha$ | 1.00 | 1.00 | 0.99 | 1.00 | 0.98 | 0.99 | 1.00 | 0.99 | 0.98 | 1.00 | 1.00 | 0.98 |
| $\beta$ | 0.99 | 0.99 | 0.98 | 0.99 | 0.99 | 0.99 | 1.00 | 1.00 | 0.97 | 1.00 | 1.00 | 0.99 |

Keenahan et al. [27] and Xue et al. [29] proposed similar methods to estimate road unevenness. The estimation accuracy of the vehicle parameters and road unevenness of their schemes can be the same as the PRE method. However, the obtained accuracy is enough for evaluating road roughness, while a higher accuracy is required for estimating bridge parameters. According to the field test results [32], the PRE accuracy is decreased by the noise. Thus, it is necessary to investigate the noise characteristics of VBI systems to apply the PRE method practically. Xue et al. [29] installed only a few sensors on vehicle bodies, while the PRE method needs more sensors at the sprung-mass and unsprung-mass.

## 5. Conclusions

This paper presents the theoretical framework of the PRE (numerical simulation-based vehicle and bridge parameter and road roughness estimation) approach, which uses the vehicle's vibration and position data to simultaneously estimate the characteristics of the vehicle and the bridge systems and the road unevenness. Additionally, it quantitatively confirms the PRE method's suitability for estimating vehicle and bridge parameters. In the numerical simulation, a half-car model and an Euler–Bernoulli beam are used as the vehicle and bridge models. Several stiffness patterns are prepared for the bridge model, and it is suggested that stiffness reduction of bridges can be detected in all scenarios. It is also demonstrated that accurately determining the extent of stiffness reduction is challenging. The PRE's uniqueness compared to existing methods [26–29] is the applicability of bridge identification. The purpose of previous schemes [26–28] is to estimate road profiles, while this study focuses on a new drive-by bridge monitoring technology, which can estimate the spatial distribution of bridge flexural rigidity and other characteristics as current performance. It is of value both academically and practically.

The noise in the observed data is not considered in this investigation. The PRE approach is probably not robust because it uses the process of numerical integration. It is necessary enhance the model to account for noise. The vehicle adopted in this study is only 10 tons, while the traffic-induced bridge vibration is not relatively dominant in the input profile. As a result, while modifying the vehicle weight and bridge span, it is also essential to consider a suitable vehicle–bridge combination for the PRE method.

The obtained accuracy for bridge damage identification is limited. The MCMC method is inefficient and overestimates the reduced stiffness of the bridge, which can lead to dangerous accidents. The reason for low estimation accuracy lies in the low efficiency of the optimization method. Therefore, the efficiency of the optimization process must be improved. The model accuracy, such as the 3D vehicle with engine vibration and 3D bridge interaction model, should also be examined.

In this study area, a field experiment is still a technical issue. When doing PRE-based bridge monitoring, the sensor installation is only required on a vehicle. However, while testing the PRE method, vibration sensors should also be installed on bridges to examine the PRE accuracy. The accuracy can be evaluated by comparing the bridge responses estimated by the PRE method with the measured data. In addition, these practical challenges have not yet been overcome. According to the previous study performing the field test [32] of the PRE method, many of the causes come from the low robustness due to observation noises. The numerical integration process is susceptible to noise, and rather than the mechanical parameters themselves, the results can be significantly influenced by unknown factors such as the angle of sensor installation, electromagnetic waves, speed fluctuation, engine vibration, and wind loads. In a field-experimental verification in the future, an actual bridge having stiffness reduction should also be prepared to discuss the estimation accuracy of reduced stiffness of bridge parts. However, bridges generally have complicated structures with spatially different stiffness. Thus, it is required to distinguish between intentionally-designed and aging-caused spatial stiffness reduction from the results. This means that the numerical simulation models must be improved.

Despite these many difficulties, the PRE method is still attractive as a bridge screening technique because it is mechanically straightforward and satisfies the challenging requirements of bridge maintenance. This study increases the feasibility of the PRE approach for actual bridge inspection, despite many remaining technical issues.

**Author Contributions:** Conceptualization, K.Y. and R.S.; methodology, K.Y.; validation, K.Y.; resources, K.Y.; data curation, K.Y. and R.S.; writing-original draft preparation, K.Y., R.S., and E.M.; writing-review and editing, K.Y.; visualization, K.Y.; supervision, K.Y.; funding acquisition, K.Y. All authors have read and agreed to the published version of the manuscript.

**Funding:** This study is supported by JSPS KAKENHI (Grant Number: JP19H02220).

**Institutional Review Board Statement:** Not applicable.

**Informed Consent Statement:** Not applicable.

**Data Availability Statement:** Data is contained within the article.

**Conflicts of Interest:** The authors declare no conflict of interest.

## Appendix A

The obtained vehicle accelerations in Scenario 1, shown in Figure 5e,f, are separately shown in Figures A1 and A2, respectively.

The enlarged views of Figures 6 and 7 are shown in Figure A3.

Figure A4 shows the differences between correct and estimated road unevenness functions. According to these figures, the estimation errors are less than 0.1 mm in most positions.

The histograms included in Figures 13–16 are enlarged in Figures A5–A14 for the convenience of the readers.

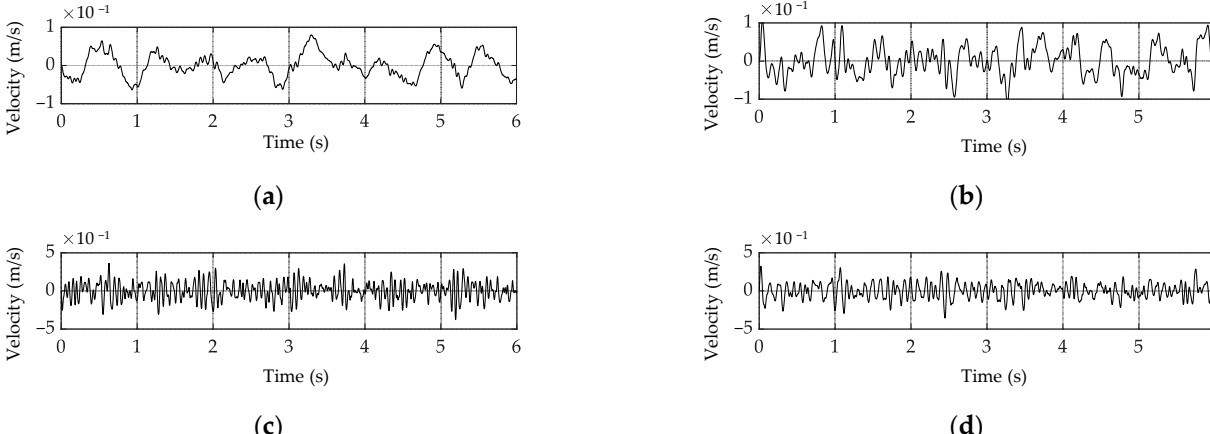

**Figure A1.** The velocity responses obtained in Scenario 1: (**a**) $\dot{z}_{s1}$; (**b**) $\dot{z}_{s2}$; (**c**) $\dot{z}_{u1}$; and (**d**) $\dot{z}_{u2}$.

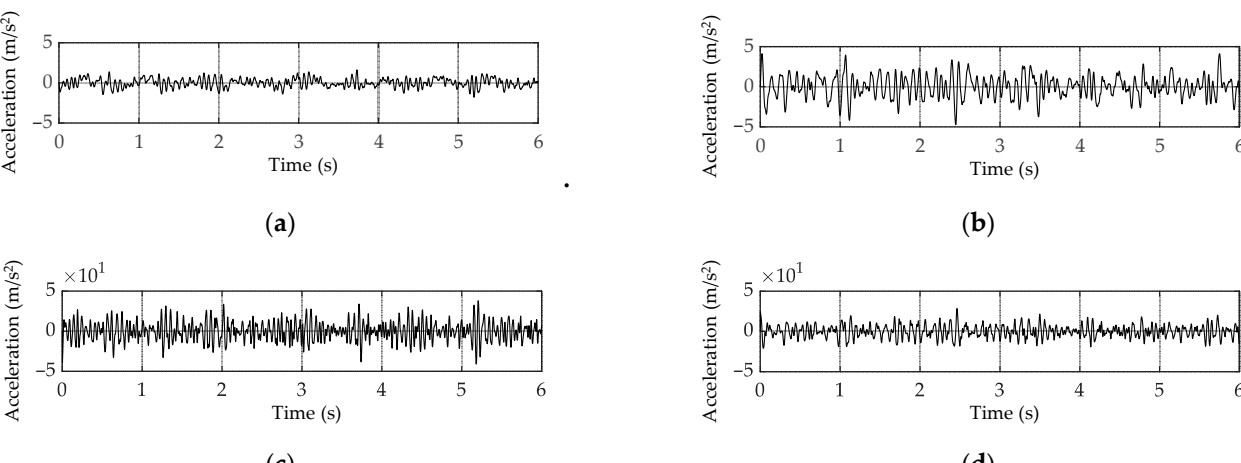

**Figure A2.** The acceleration responses obtained in Scenario 1: (**a**) $\ddot{z}_{s1}$; (**b**) $\ddot{z}_{s2}$; (**c**) $\ddot{z}_{u1}$; and (**d**) $\ddot{z}_{u2}$.

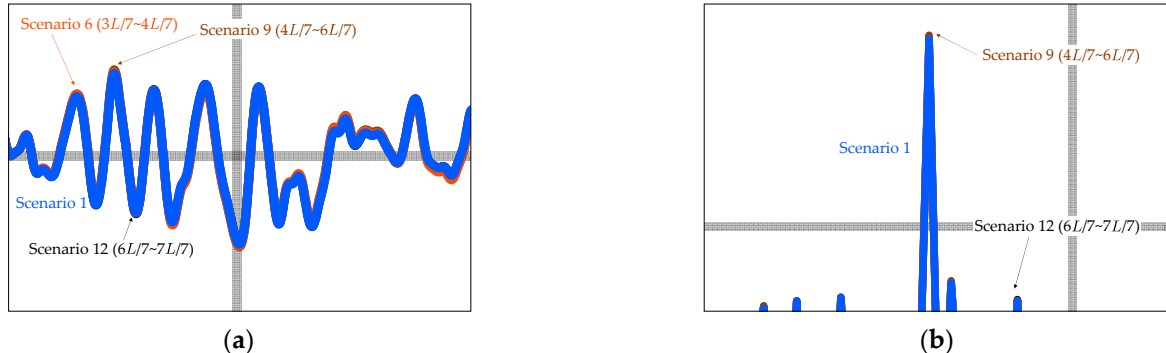

**Figure A3.** Enlarged view of Figure 6a $\ddot{z}_{s1}(t)$ and Figure 7c showing the power spectrum of $\ddot{z}_{u1}$: (**a**) Figure 6; (**b**) Figure 7. The differences in the vehicle responses between different scenarios are tiny because the influence of fixed road profiles is predominant.

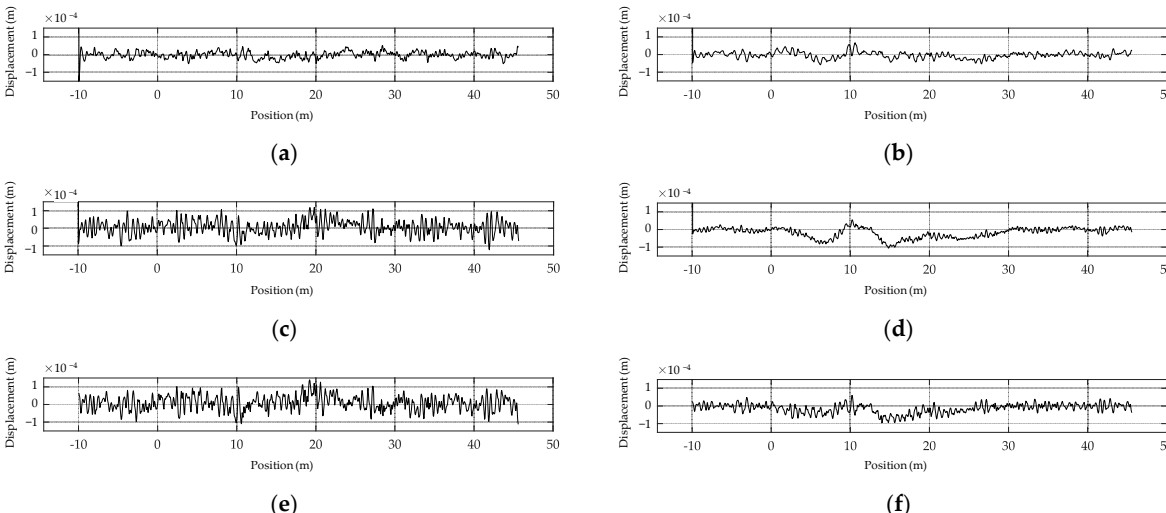

**Figure A4.** The residual of the correct/estimated road unevenness shown in Figures 10 and 12: (**a**) $\hat{R}_1 - \hat{R}_2$ in Scenario 1; (**b**) $\hat{R}_1 - \hat{R}_2$ in Scenario 6; (**c**) $\hat{R}_1 - R$ in Scenario 1; (**d**) $\hat{R}_1 - R$ in Scenario 6; (**e**) $\hat{R}_2 - R$ in Scenario 1; and (**f**) $\hat{R}_2 - R$ in Scenario 6.

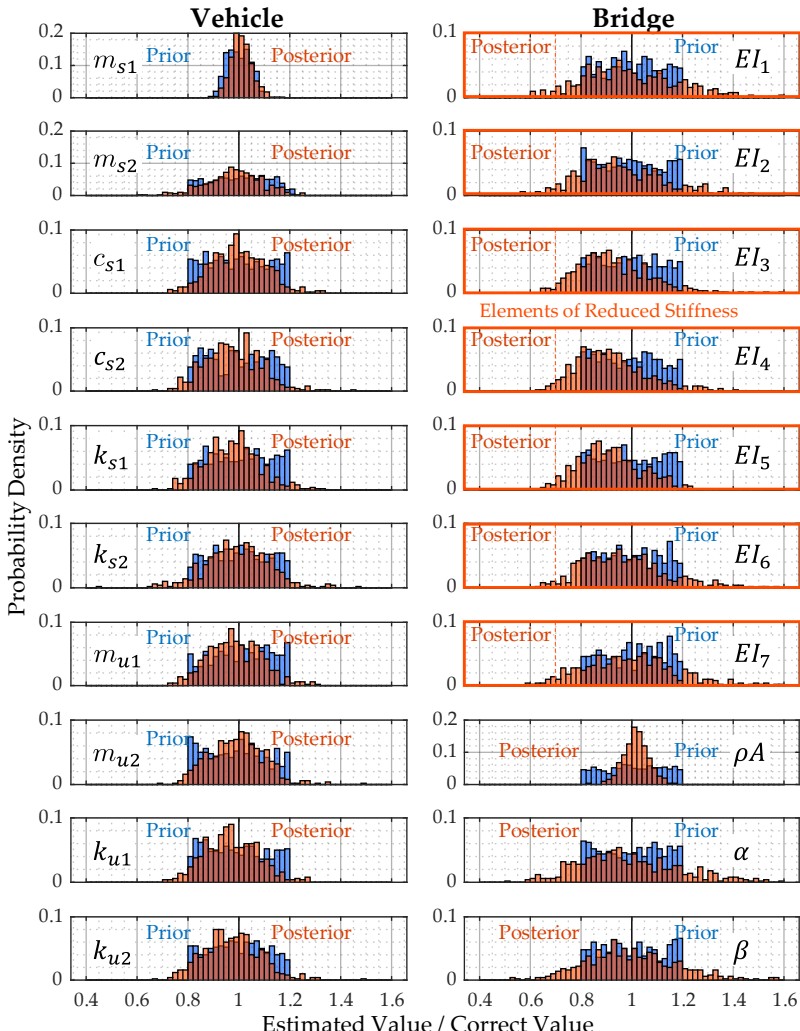

**Figure A5.** The application results of the PRE method in the case of Scenario 2 (stiffness decrease section: 0~$L$, decrease rate: 10%).

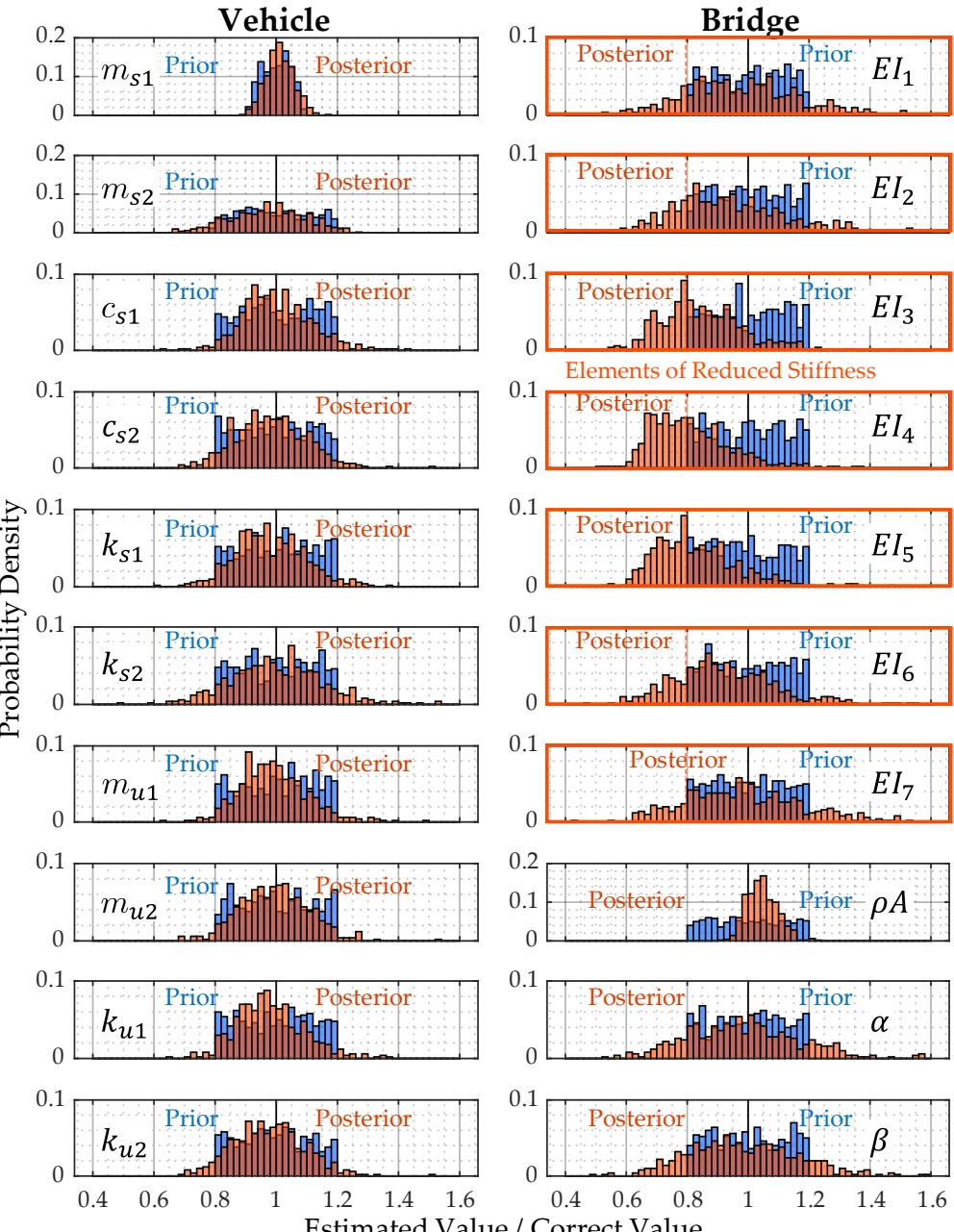

**Figure A6.** The application results of the PRE method in the case of Scenario 3 (stiffness decrease section: 0~*L*, decrease rate: 20%).

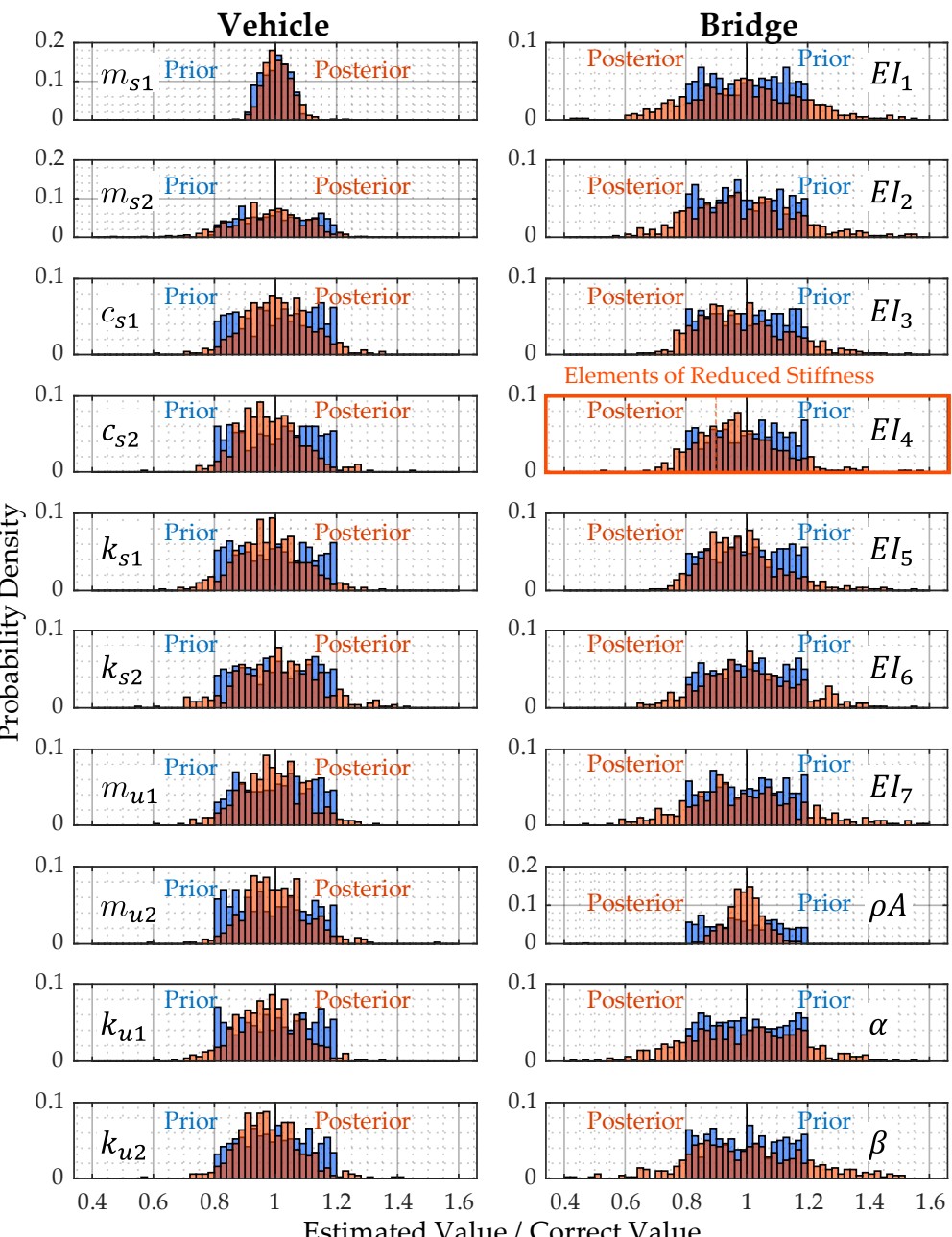

**Figure A7.** The application results of the PRE method in the case of Scenario 4 (stiffness decrease section: 3*L*/7~4*L*/7, decrease rate: 10%).

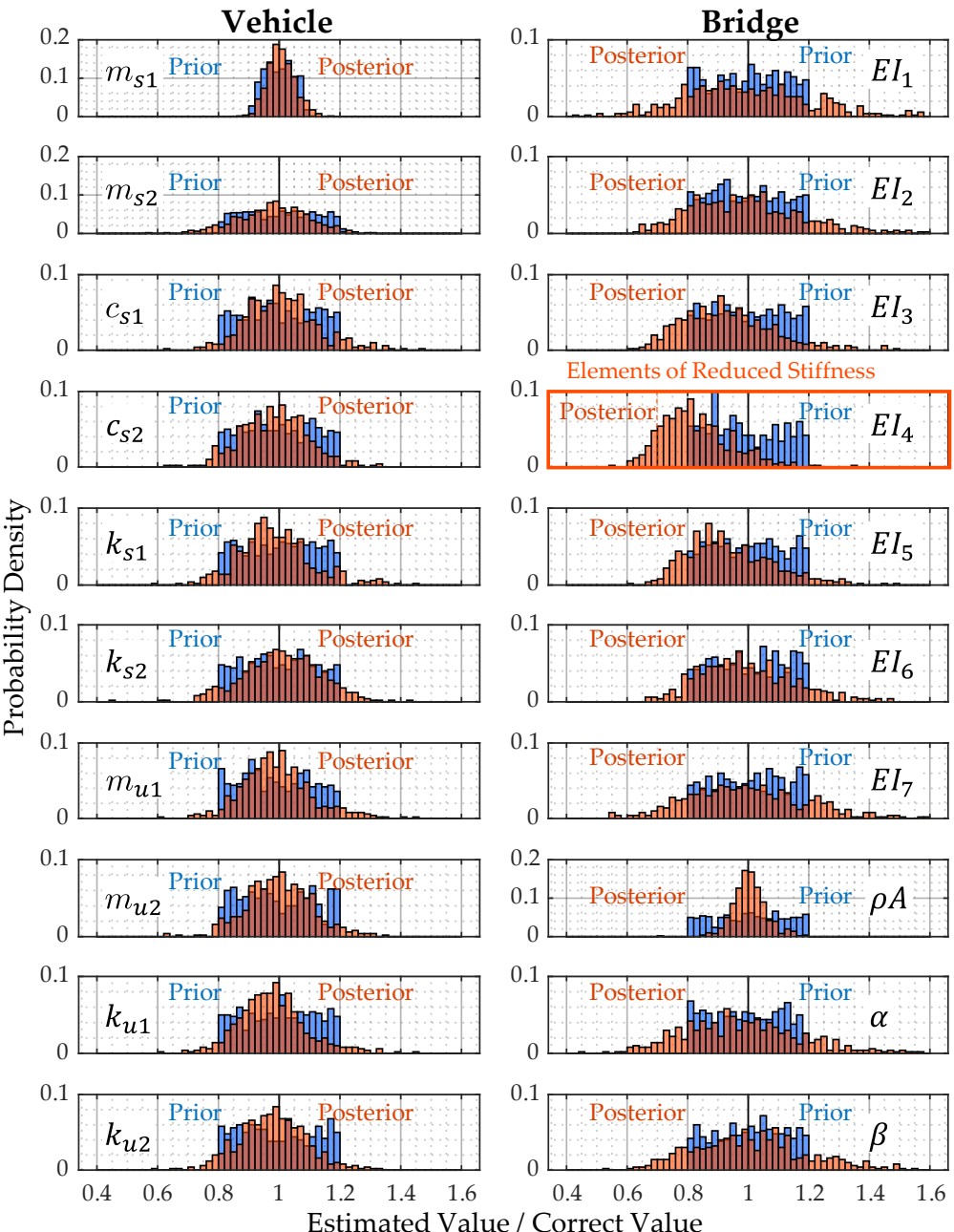

**Figure A8.** The application results of the PRE method in the case of Scenario 5 (stiffness decrease section: $3L/7\sim4L/7$, decrease rate: 30%).

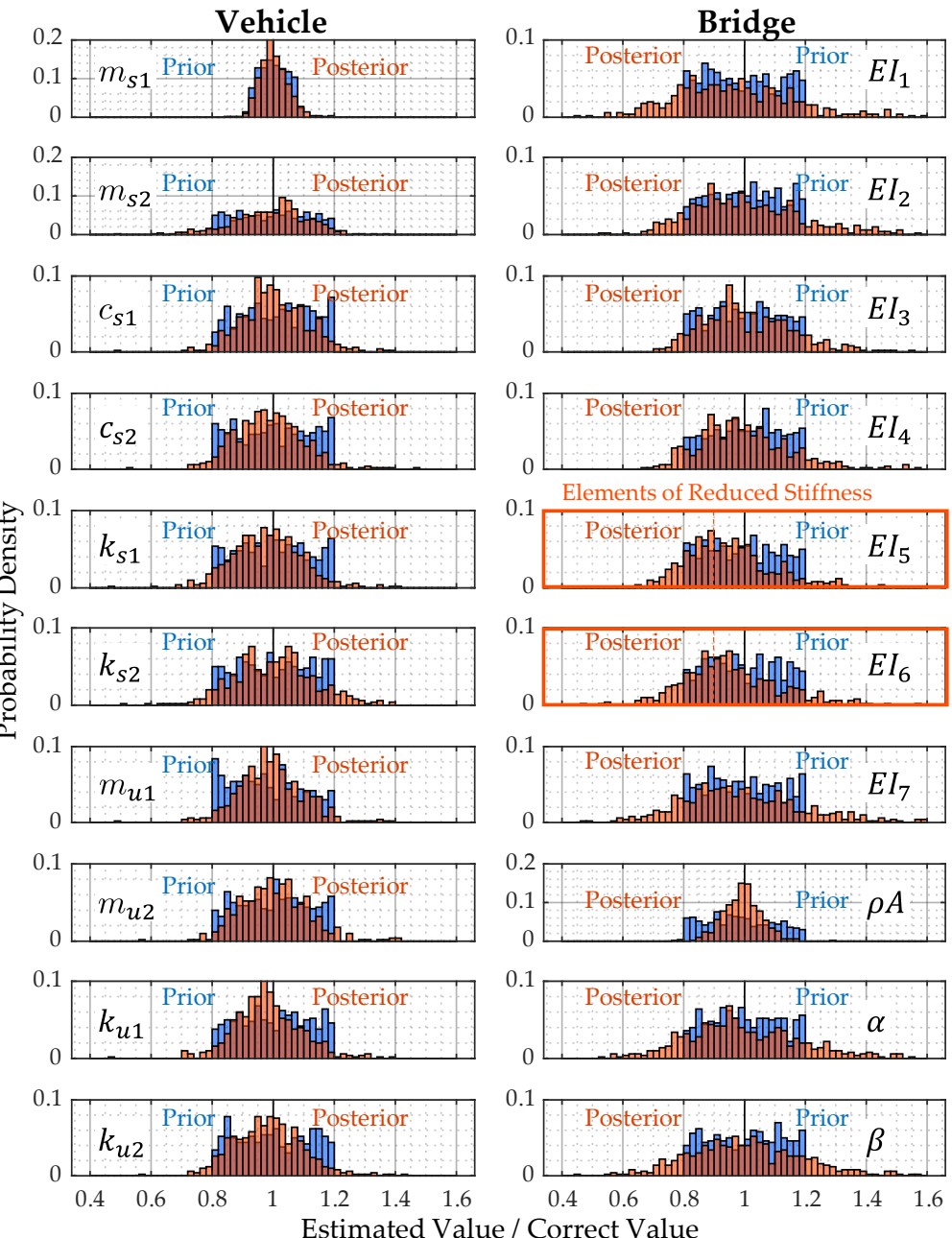

**Figure A9.** The application results of the PRE method in the case of Scenario 7 (stiffness decrease section: 4*L*/7~6*L*/7, decrease rate: 10%).

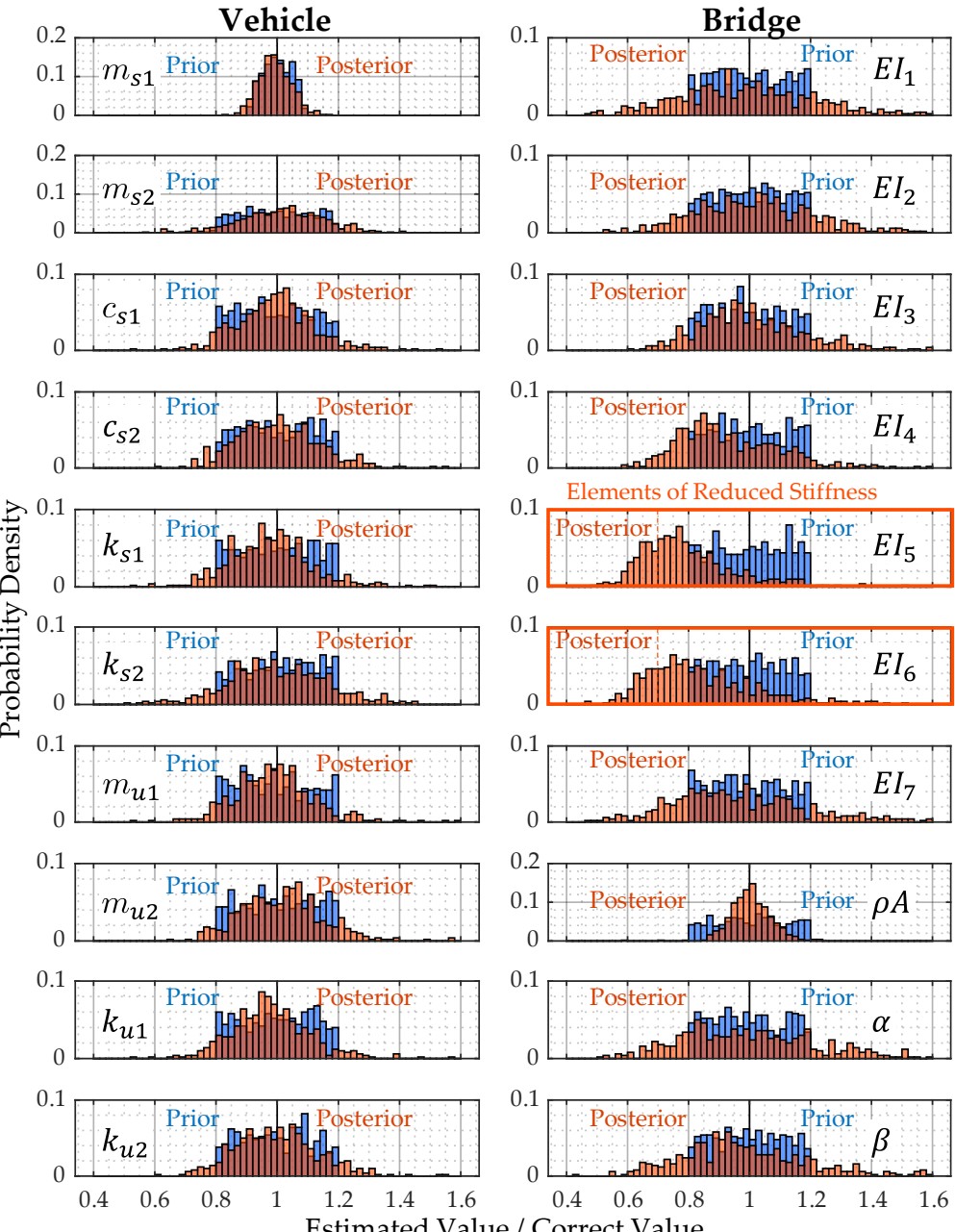

**Figure A10.** The application results of the PRE method in the case of Scenario 8 (stiffness decrease section: $4L/7\sim6L/7$, decrease rate: 30%).

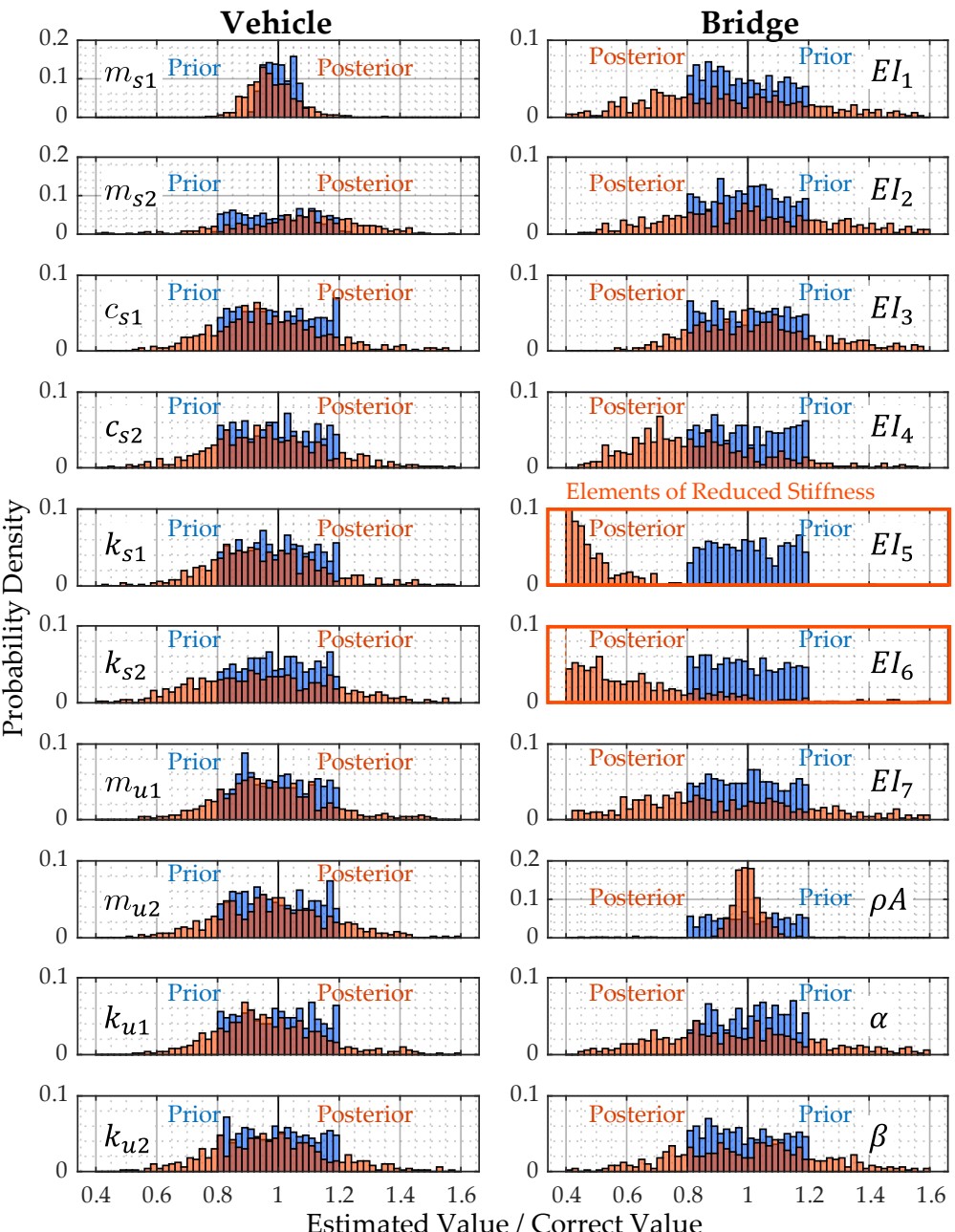

**Figure A11.** The application results of the PRE method in the case of Scenario 9 (stiffness decrease section: 4*L*/7~6*L*/7, decrease rate: 60%).

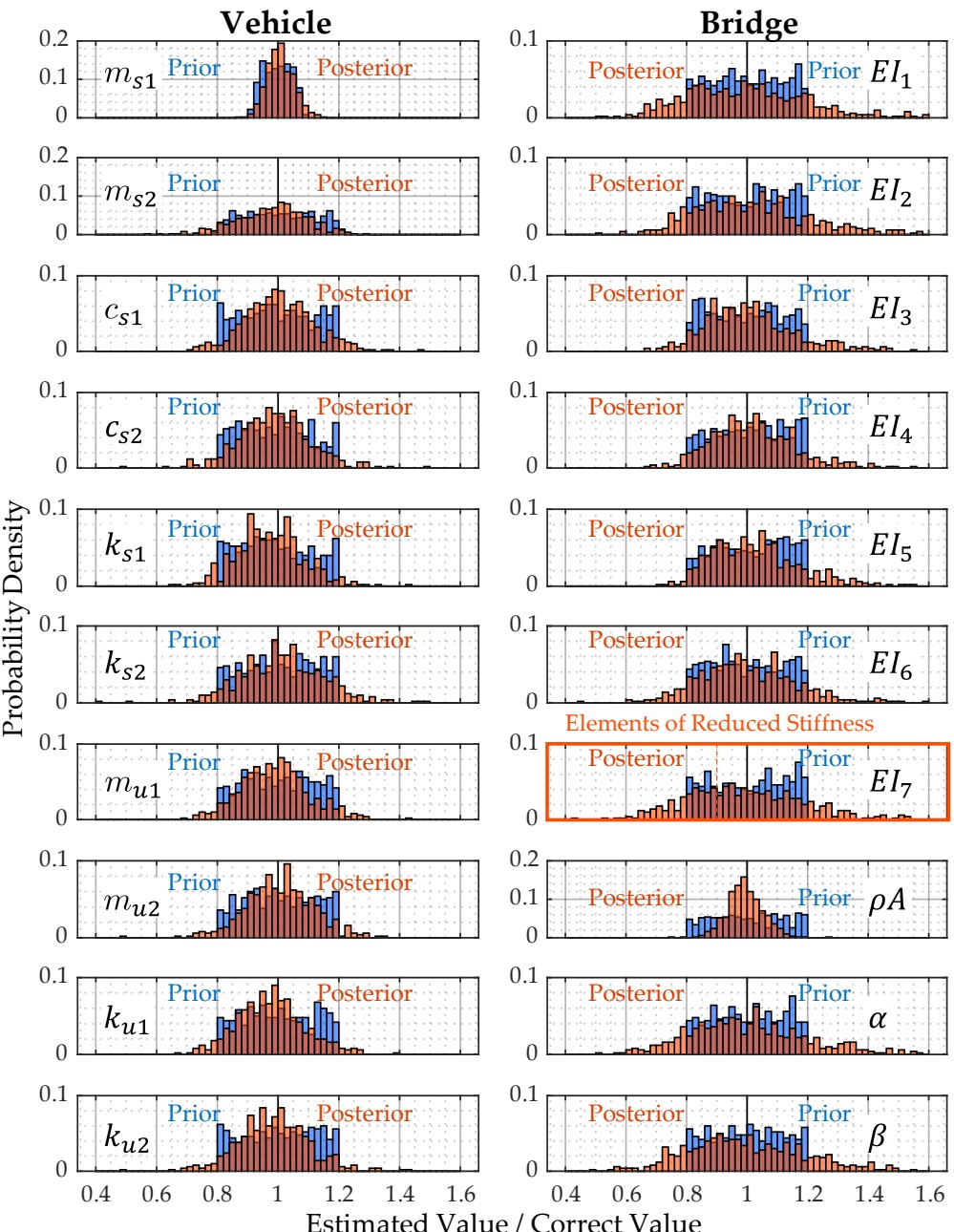

**Figure A12.** The application results of the PRE method in the case of Scenario 10 (stiffness decrease section: $6L/7 \sim L$, decrease rate: 10%).

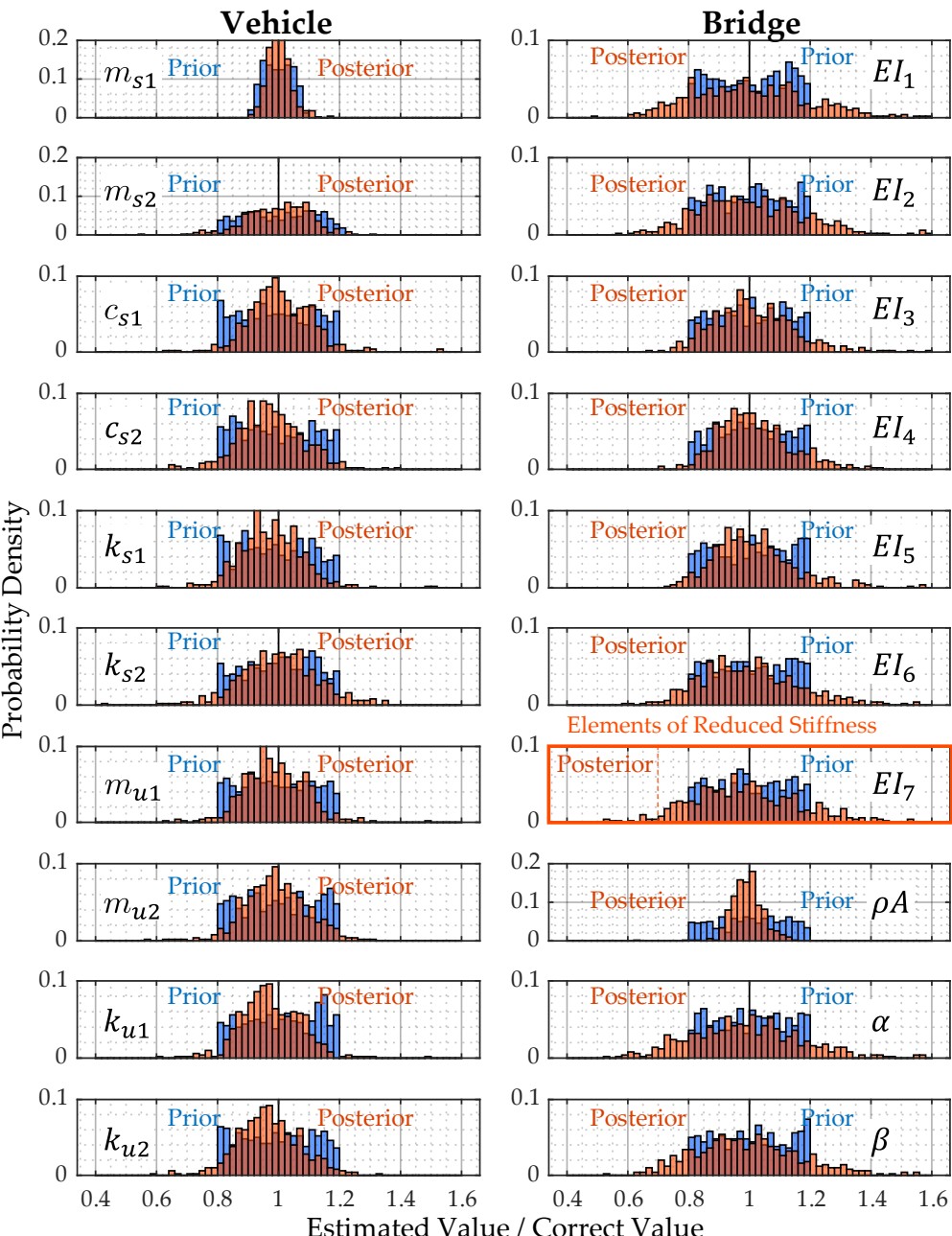

**Figure A13.** The application results of the PRE method in the case of Scenario 11 (stiffness decrease section: $6L/7 \sim L$, decrease rate: 30%).

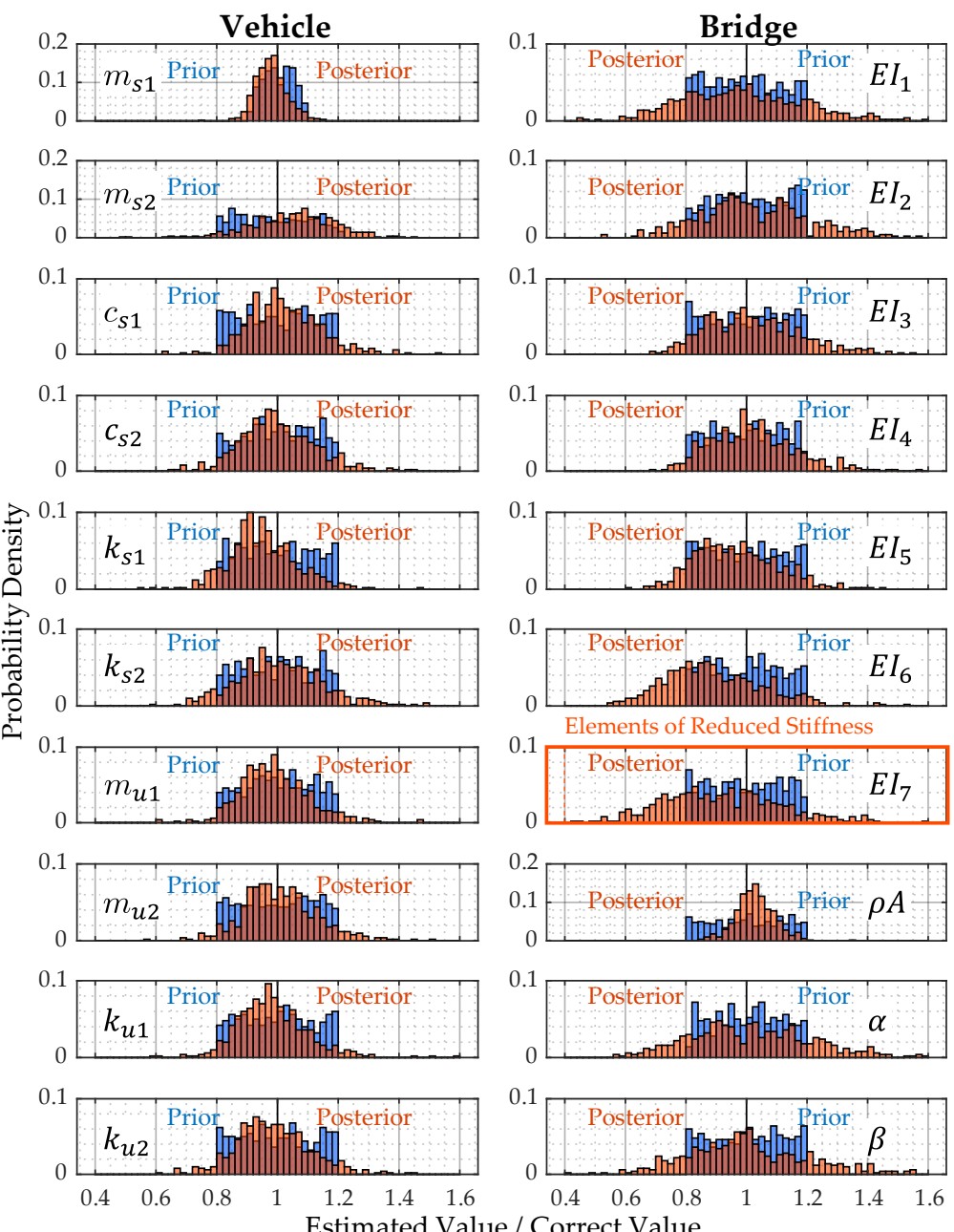

**Figure A14.** The application results of the PRE method in the case of Scenario 12 (stiffness decrease section: $6L/7\sim L$, decrease rate: 60%).

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
