# Peer review of "Numerical Verification of the Drive-By Monitoring Method for Identifying Vehicle and Bridge Mechanical Parameters"

_applsci, doi:10.3390/app13053049_

Round 1

Reviewer 1 Report

Review: Applicability of the Vehicle-Bridge Interaction System Identification (VBISI) Method to Bridge Damage Estimation

The title is confusing. It is better to change it. Vehicle Bridge Interaction System (VBIS) is a field of study that has been discussed in the past. However, the term “VBISI method” does not make any sense. What does it mean? Either clarify the meaning of this term or change it to VBIS or its suitable variant (e.g. numerical simulation-based VBIS, Markov model-based VBIS).

The title also contains the keyword “bridge damage estimation”. However, no evidence has been found with respect to damage estimation. This is a tall claim, which involves actual utilization of the proposed method for damage estimation of an actual bridge. However, no such data has been used. Numerical simulation has been used, but the authors did not mention how they were calculating the performance of the numerical simulation of proposed method. Is the proposed method able to accurately estimate damage? Is it better than the state-of-the-art? These questions remain unanswered throughout the manuscript.

Furthermore, when you talk about “damage estimation”, you have to rely on existing metrics for damage estimation that are widely used in the existing literature pertaining to bridges. What metrics have been used for damage assessment? What is the basis for using those metrics? Are those metrics verified and validated in the existing literature?

1. Introduction

In-text citation style is not consistent. Numerical citation has been used. Year information has also been provided, which is not required. The last name of author has to be used with the respective citation numbering, instead full name of authors has been highlighted.

26-27 studies are discussed as part of the background. However, each study is discussed independently, but no synthesis or connection has been made from one study to another. So, instead of an overview of the different studies, what is currently given is sentences talking about individual studies, but there is no link established between individual studies to provide a holistic overview of the existing research landscape.

Furthermore, the authors have not been very clear with respect to highlighting how the existing work fits into and extends/improves the state-of-the-art? This point needs to be clearly highlighted and justified in this section.

2. Numerical Simulation of Vehicle-Bridge Interaction System

The matrices in eq. (2), eq. (3), eq. (4), eq. (17), eq. (18), and eq. (19) have empty spaces that should ideally contain some variables or constants. No explanation has been given to explain this phenomena. This needs to be rectified.

I do not understand the rationale behind “random assumption of vehicle and bridge mechanical properties”. This needs to be justified. How did you come up with this technique? Is this technique verified and validated in the relevant literature? If not, then clearly justify why and how this is an appropriate method that is practical for bridge damage assessment.

How did you come up with values in table 1 and 2? Justify how and why did you come up with these values?

In table 2, there are columns about damage 1, damage 2, and damage 4. Column about damage 3 is missing.

In table 2, how did you make calculations regarding damage 1, damage 2 and damage 4? If there is some theoretical or practical evidence to highlight this point. It should be highlighted in this section. There is some discussion between lines 249-255. However, currently, it is unclear how these calculations were performed and what is the scientific evidence to support your assertions?

How did you come up with the calculations in table 4? Justify this point clearly.

3. The Vehicle-Bridge Interaction System Identification Method

Figure 8 is unclear. Please remake figure 8 and try to break down the vehicle and bridge input estimation blocks into further sub-blocks that make the figure more clear. The different steps for calculation should be made more clear.

The sub-figures in Figure 9 are too small and it is difficult for the reader to view and examine what is being highlighted. The size of individual sub-figures should be increased. Each case should have a separate figure. For example, sub-figure (a) for intact case should be figure 9. sub-figure (b) should be figure 10 and so on.

Size of figure 10 should be increased to clearly highlight variation between estimated front and rear road roughness and actual road roughness. This aspect is unclear in figure 10.

This section should have clearly highlighted and discussed the actual performance of the proposed VBISI method. No such information or statistics have been highlighted. Figure 9 highlights the application results of VBISI method with the estimated and correct values. What is the percentge error between the estimated and correct values for the different cases? Four cases are insufficient to provide sufficient evidence regarding the performance of VBISI method. At least ten different damage scenarios should have been highlighted and their performance should have been examined in this section. Furthermore, the authors should have compared the performance of the VBISI method with the existing works. Is the performance better or comparable to the state-of-the-art? This point needs to be clearly discussed in this section.

Reviewer 2 Report

This paper proposed a VBISI method that simultaneously identifies the vehicle mechanical parameters, bridge pavement roughness, and possible bridge stiffness deterioration. Basic theories of vehicle-bridge interaction are well introduced. Numerical results show that it is possible to extracts bridge roughness and deterioration from vehicle vibration data. Some suggestions are below.

1. The organization of this paper could be further improved. For example, both Section 2 and Section 3 has a sub-section named “numerical simulation”. However, they correspond to the forward and inverse problem. The logic of the text as well as the titles and sub-titles could be considered more carefully.

2. Figure 9 may need further explanations. The posterior distribution is estimated instead of a deterministic value? In the current manner, how is the estimation accuracy measured? An index that quantitatively evaluates the estimation results is strongly desired.

3. Some other minor errors: In Table 3, “Damage 4” seems be wrongly typed as “Damage 3”. Line 361, “The vertical axis does the frequency” is incomprehensible. Line 435, what does “It is a dangerous result” indicate?

Round 2

Reviewer 1 Report

Comments 1-20 belong to the original review (i.e. Review 1) that was submitted earlier. The reviewer will first examine whether the original comments have been addressed or not. Then, mentioning this point and further clarify. Comments 21 onwards are related to the supposed improvements and additions in the updated manuscript that relate to Review 2.

Comment 1 in Review 1: The title is confusing. It is better to change it. Vehicle Bridge Interaction System (VBIS) is a field of study that has been discussed in the past. However, the term “VBISI method” does not make any sense. What does it mean? Either clarify the meaning of this term or change it to VBIS or its suitable variant (e.g. numerical simulation-based VBIS, Markov model-based VBIS).

Comment 1 in Review 2: The title name has been changed. However, the name of the proposed method remains VBISI and the authors do not provide any justification in the paper to explain why this method is called VBISI. Hence, this comment has not been fully addressed by the authors.

Comment 2 in Review 1: The title also contains the keyword “bridge damage estimation”. However, no evidence has been found with respect to damage estimation. This is a tall claim, which involves actual utilization of the proposed method for damage estimation of an actual bridge. However, no such data has been used.

Comment 2 in Review 2: This comment has also not been addressed by the authors. By mentioning “bridge damage estimation”, the authors need to understand that justifying this keyword would require the addition of actual bridge data in the manuscript and develop a thorough comparison of the numerical simulation performed in this manuscript with the actual bridge damage data to effectively highlight the efficacy of the proposed method. No such evidence has been provided. Therefore, the use of this phrase “ should be avoided and replaced with some other more suitable alternate (e.g. use of numerical simulation for measuring bridge surface unevenness, numerical simulation-based bridge and vehicle parameter estimation in VBI systems).

Comment 3 in Review 1: Numerical simulation has been used, but the authors did not mention how they were calculating the performance of the numerical simulation of proposed method. Is the proposed method able to accurately estimate damage? Is it better than the state-of-the-art? These questions remain unanswered throughout the manuscript.

Comment 3 in Review 2: Some improvements can be seen with respect to mentioning how the different calculations have been made for the different parameters. The authors have also added a table highlighting the accuracy of parameter estimation. However, there is no comparison provided with the performance of the existing works. Therefore, this point has not been improved by the authors.

Comment 4 in Review 1: Furthermore, when you talk about “damage estimation”, you have to rely on existing metrics for damage estimation that are widely used in the existing literature pertaining to bridges. What metrics have been used for damage assessment? What is the basis for using those metrics? Are those metrics verified and validated in the existing literature?

Comment 4 in Review 2: This point has been, for the most part, been addressed by the authors.

1. Introduction

Comment 5 in Review 1: In-text citation style is not consistent. Numerical citation has been used. Year information has also been provided, which is not required. The last name of author has to be used with the respective citation numbering, instead full name of authors has been highlighted.

Comment 5 in Review 2: This point has not been addressed by the authors. The manner in which in-text citation has been performed is still incorrect. For example, in line 77, the in-text citation is “Y. B. Yang et al. [15] …”. This needs to be changed to “Yang et al. [15] …”. Similarly, in line 84, the in-text citation used is: “Y. B. Yang and K. C. Chang [17] …”. This should be changed to the following: “Yang and Chang [17] …”. In the similar fashion, all other citations should be modified throughout the paper.

Comment 6 in Review 1: 26-27 studies are discussed as part of the background. However, each study is discussed independently, but no synthesis or connection has been made from one study to another. So, instead of an overview of the different studies, what is currently given is sentences talking about individual studies, but there is no link established between individual studies to provide a holistic overview of the existing research landscape.

Comment 6 in Review 2: This point has not been addressed by the authors. Instead, the authors have tried to add more studies in the literature review, which is fine. However, this point should be rectified.

Comment 7 in Review 1: Furthermore, the authors have not been very clear with respect to highlighting how the existing work fits into and extends/improves the state-of-the-art? This point needs to be clearly highlighted and justified in this section.

Comment 7 in Review 2: This point has been improved by the authors.

Comment 21 in Review 2: Upon reviewing and reading some of the papers given in the literature review of the manuscript, the authors fail to mention that the existing works (e.g. Zhao et al., 2019) utilize data from actual vehicles and compare and validate the numerical simulation results with data acquired from actual experiments involving data collection from different vehicles on actual roads/bridges. Not including actual data from experimentation on real bridges and vehicles, especially when the existing works have already incorporated this data is a major drawback of this manuscript. This issue needs to be addressed by the authors.

2. Numerical Simulation of Vehicle-Bridge Interaction System (New redefined as “Numerical Simulation”)

Comment 8 in Review 1: The matrices in eq. (2), eq. (3), eq. (4), eq. (17), eq. (18), and eq. (19) have empty spaces that should ideally contain some variables or constants. No explanation has been given to explain this phenomena. This needs to be rectified.

Comment 8 in Review 2: This point has been rectified by the authors.

Comment 9 in Review 1: I do not understand the rationale behind “random assumption of vehicle and bridge mechanical properties”. This needs to be justified. How did you come up with this technique? Is this technique verified and validated in the relevant literature? If not, then clearly justify why and how this is an appropriate method that is practical for bridge damage assessment.

Comment 9 in Review 2: This point has not been justified by the authors.

Comment 10 in Review 1: How did you come up with values in table 1 and 2? Justify how and why did you come up with these values?

Comment 10 in Review 2: This point has not been justified by the authors.

Comment 11 in Review 1: In table 2, there are columns about damage 1, damage 2, and damage 4. Column about damage 3 is missing.

Comment 11 in Review 2: This point has been rectified by the authors.

Comment 12 in Review 1: In table 2, how did you make calculations regarding damage 1, damage 2 and damage 4? If there is some theoretical or practical evidence to highlight this point. It should be highlighted in this section. There is some discussion between lines 249-255. However, currently, it is unclear how these calculations were performed and what is the scientific evidence to support your assertions?

Comment 12 in Review 2: The authors have increased the quantity of damaged cases being discussed. However, the issue highlighted in this point has still not been clearly highlighted. If anything, this confusion has been increased. For global damage and local damage simulation, how are you changing the values for the different variables? Mention the original source in which this has been mentioned to validate the method used for changing variables for local and global damage simulation.

Comment 13 in Review 1: How did you come up with the calculations in table 4? Justify this point clearly.

Comment 13 in Review 2: This point has been justified by the authors to some extent.

Comments 22 in Review 2: The graphs in figure 5, 6, and 7 are too small in size to appreciate the difference in the values between the different graph plots. Increase the size of each of the sub-figures.

3. The Vehicle-Bridge Interaction System Identification Method (Now redefined as “The Proposed Method”)

Comment 14 in Review 1: Figure 8 is unclear. Please remake figure 8 and try to break down the vehicle and bridge input estimation blocks into further sub-blocks that make the figure more clear. The different steps for calculation should be made more clear.

Comment 14 in Review 2: This point has been rectified by the authors.

Comment 15 in Review 1: The sub-figures in Figure 9 are too small and it is difficult for the reader to view and examine what is being highlighted. The size of individual sub-figures should be increased. Each case should have a separate figure. For example, sub-figure (a) for intact case should be figure 9. sub-figure (b) should be figure 10 and so on.

Comment 15 in Review 2: This point has been improved by the authors.

Comment 16 in Review 1: Size of figure 10 should be increased to clearly highlight variation between estimated front and rear road roughness and actual road roughness. This aspect is unclear in figure 10.

Comment 16 in Review 2: Figure 10 is still too small to see how the estimated and actual values for the front and rear wheels differ. This point has not been rectified by the authors.

Comment 17 in Review 1: This section should have clearly highlighted and discussed the actual performance of the proposed VBISI method. No such information or statistics have been highlighted.

Comment 17 in Review 2: This point has been improved by the authors.

Comment 18 in Review 1: Figure 9 highlights the application results of VBISI method with the estimated and correct values. What is the percentage error between the estimated and correct values for the different cases?

Comment 18 in Review 2: This point has been improved by the authors.

Comment 19 in Review 1: Four cases are insufficient to provide sufficient evidence regarding the performance of VBISI method. At least ten different damage scenarios should have been highlighted and their performance should have been examined in this section.

Comment 19 in Review 2: This point has been addressed by the authors.

Comment 20 in Review 1: Furthermore, the authors should have compared the performance of the VBISI method with the existing works. Is the performance better or comparable to the state-of-the-art? This point needs to be clearly discussed in this section.

Comment 20 in Review 2: This point has not been addressed by the authors.

Comment 23 in Review 2: Figure 12-16 are too small and the size is preventing from actually seeing how the actual and estimated values differ for each of the damaged cases.

Comment 24 in Review 2: Table 6 is redundant and should be removed. Mentioning the average and standard deviation values for the different parameters does not tell us much about the performance of VBISI method.

Comment 25 in Review 2: The estimation accuracy has been highlighted in table 7. Why are some values given in bold. Mention that in the text.

5. Conclusion

Comment 26 in Review 2: Line 550-551 should be changed or deleted. There is no evidence to suggest the suitability of VBISI for estimating bridge damage. At best, using numerical simulation data, the VBISI has been used for vehicle and bridge parameter estimation.

Comment 27 in Review 2: Line 555-556 should be changed or deleted entirely. VBISI method outlined in this paper has not been used for bridge maintenance or bridge damage estimation. In order to validate this statement, the authors need to include data from actual experiments involving actual cars, actual bridges and actual sensors. Currently, there is no evidence of using data from actual experiments in this manuscript.

Comment 28 in Review 2: Between line 575-585, the authors highlight the challenges with field experiments. That is expected. However, existing works use data from field experiments (e.g. Zhao et al., 2018, 2019). Therefore, in order to validate the efficacy of VBISI and compare it to existing works, data from field experiments should be added in the manuscript.
